# TRACKING THE RISK OF A DEPLOYED MODEL AND DETECTING HARMFUL DISTRIBUTION SHIFTS

**Aleksandr Podkopaev**[1,2]**, Aaditya Ramdas**[1,2]
Department of Statistics & Data Science[1]
Machine Learning Department[2]
Carnegie Mellon University
{podkopaev,aramdas}@cmu.edu

## ABSTRACT

When deployed in the real world, machine learning models inevitably encounter changes in the data distribution, and certain—but not all—distribution shifts could result in significant performance degradation. In practice, it may make sense to ignore benign shifts, under which the performance of a deployed model does not degrade substantially, making interventions by a human expert (or model retraining) unnecessary. While several works have developed tests for distribution shifts, these typically either use non-sequential methods, or detect arbitrary shifts (benign or harmful), or both. We argue that a sensible method for firing off a warning has to both (a) detect harmful shifts while ignoring benign ones, and (b) allow continuous monitoring of model performance without increasing the false alarm rate. In this work, we design simple sequential tools for testing if the difference between source (training) and target (test) distributions leads to a significant increase in a risk function of interest, like accuracy or calibration. Recent advances in constructing time-uniform confidence sequences allow efficient aggregation of statistical evidence accumulated during the tracking process. The designed framework is applicable in settings where (some) true labels are revealed after the prediction is performed, or when batches of labels become available in a delayed fashion. We demonstrate the efficacy of the proposed framework through an extensive empirical study on a collection of simulated and real datasets.

## 1 INTRODUCTION

Developing a machine learning system usually involves data splitting where one of the labeled folds is used to assess its generalization properties. Under the assumption that the incoming test instances (target) are sampled independently from the same underlying distribution as the training data (source), estimators of various performance metrics, such as accuracy or calibration, are accurate. However, a model deployed in the real world inevitably encounters variability in the input distribution, a phenomenon referred to as *dataset shift; see the book by Quionero-Candela et al. (2009). Commonly studied settings* include covariate shift (Shimodaira, 2000) and label shift (Saerens et al., 2002). While testing whether a distribution shift is present has been studied both in offline (Rabanser et al., 2019; Gretton et al., 2012; Hu & Lei, 2020) and online (Vovk et al., 2005; Vovk, 2020; 2021) settings, a natural question is whether an intervention is required once there is evidence that a shift has occurred.

A trustworthy machine learning system has to be supplemented with a set of tools designed to raise alarms whenever critical changes to the environment take place. Vovk et al. (2021) propose retraining once an i.i.d. assumption becomes violated and design corresponding online testing protocols. However, naively testing for the presence of distribution shift is not fully practical since it does not take into account the *malignancy* of a shift (Rabanser et al., 2019). To elaborate, users are typically interested in how a model performs according to certain prespecified metrics. In *benign* scenarios, distribution shifts could be present but may not significantly affect model performance. Raising unnecessary alarms might then lead to delays and a substantial increase in the cost of model deployment. The recent approach by Vovk et al. (2021) based on conformal test martingales is highly dependent on the choice of conformity score. In general, the methodology raises an alarm whenever

a deviation from i.i.d. is detected, which does not necessarily imply that the deviation is harmful (see Appendix A.2).

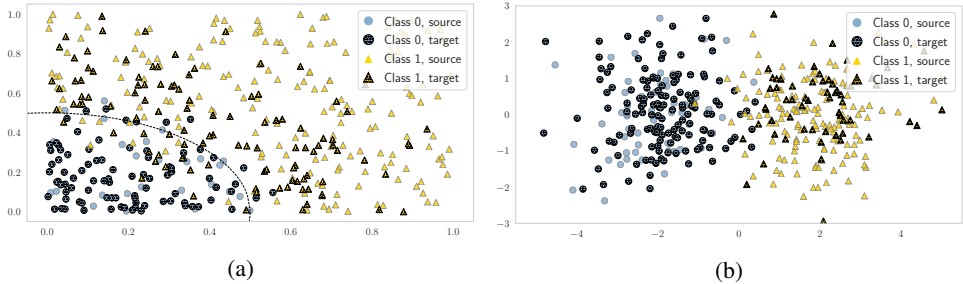

Figure 1: Samples from the source and the target (hatched) distributions under benign (a) covariate and (b) label shifts. (a) $X \sim \mathrm{Unif}([0,1] \times [0,1])$ on the source and $X_i \sim \mathrm{Beta}(1,2)$, $i = 1, 2$ on the target. Labels satisfy: $\mathbb{P}\left(Y = 1 \mid X = x\right) = \mathbb{P}\left(x_1^2 + x_2^2 + \varepsilon \geq 1/4\right)$ where $\varepsilon \sim \mathcal{N}(0, 0.01)$. (b) Marginal probability of class 1 changes from $\pi_1^S = 0.7$ on the source to $\pi_1^T = 0.3$ on the target. Covariates satisfy: $X \mid Y = y \sim \mathcal{N}(\mu_y, I_2)$, where $\mu_0 = (-2, 0)^\top$, $\mu_1 = (2, 0)^\top$. In both cases, a model which separates well data from the source will generalize well to the target.

In some cases, it is possible to handle structured shifts in a post-hoc fashion without performing expensive actions, such as model retraining. One example arises within the context of distribution-free uncertainty quantification where the goal is to supplement predictions of a model with a measure of uncertainty valid under minimal assumptions. Recent works (Tibshirani et al., 2019; Gupta et al., 2020; Podkopaev & Ramdas, 2021) show how to adapt related procedures for handling covariate and label shifts without labeled data from the target. However, both aforementioned shifts impose restrictive assumptions on the possible changes in the underlying probability distribution, assuming either that $P(X)$ changes but $P(Y|X)$ stays unchanged (covariate shift assumption), or that $P(Y)$ changes but $P(X|Y)$ stays unchanged (label shift assumption).

Thinking of distribution shifts only in terms of covariate or label shifts has two drawbacks: such (unverifiable) assumptions often may be unrealistic, and even if they were plausible, such shifts may be benign and thus could be ignored. To elaborate on the first point, it is evident that while distribution shifts constantly occur in practice, they may generally have a more complex nature. In medical diagnosis, $P(Y)$ and $P(X|Y = y)$ could describe the prevalence of certain diseases in the population and symptoms corresponding to disease $y$. One might reasonably expect not only the former to change over time (say during flu season or epidemics) but also the latter (due to potential mutations and partially-effective drugs/vaccines), thus violating both the covariate and label shift assumptions. Regarding the second point, a model capable of separating classes sufficiently well on the source distribution can sometimes generalize well to the target. We illustrate such benign covariate and label shifts on Figures 1a and 1b respectively. We argue that the critical distinction— from the point of view on raising alarms—should be built between harmful and benign shifts, and not between covariate and label shifts.

A related question is whether labeled data from one distribution can be used for training a model in a way that it generalizes well to another distribution where it is hard to obtain labeled examples. Importance-weighted risk minimization can yield models with good generalization properties on the target domain, but the corresponding statistical guarantees typically become vacuous if the importance weights are unbounded, which happens when the source distribution's support fails to cover the target support. Adversarial training schemes (Ganin et al., 2016; Wu et al., 2019) for deep learning models often yield models with reasonable performance on some types of distribution shifts, but some unanticipated shifts could still degrade performance. This paper does not deal with how to train a model if a particular type of shift is anticipated; it answers the question of when one should consider retraining (or re-evaluating) a currently deployed model.

We argue for triggering a warning once the non-regularities in the data generating distribution lead to a statistically significant increase in a user-specified risk metric. We design tools for nonparametric sequential testing for an unfavorable change in a chosen risk function of any black-box model. The

procedure can be deployed in settings where (some) true target labels can be obtained, immediately after prediction or in a delayed fashion.

During the preparation of our paper, we noticed a very recent preprint (Kamulete, 2021) on broadly the same topic. While they also advise against testing naively for the presence of a shift, their approach is different from ours as (a) it is based on measuring the malignancy of a shift based on outlier scores (while we suggest measuring malignancy via a drop in test accuracy or another pre-specified loss), and arguably more importantly (b) their procedure is non-sequential (it is designed to be performed once, e.g, at the end of the year, and cannot be continuously monitored, but ours is designed to flag alarms at any moment when a harmful shift is detected). Adapting fixed-time testing to the sequential settings requires performing corrections for multiple testing: if the correction is not performed, the procedure is no longer valid, but if naively performed, the procedure becomes too conservative, due to the dependence among the tests being ignored (see Appendix A.1). We reduce the testing problem to performing sequential estimation that allows us to accumulate evidence over time, without throwing away any data or the necessity of performing explicit corrections for multiple testing; these are implicitly handled efficiently by the martingale methods that underpin the sequential estimation procedures (Howard et al., 2021; Waudby-Smith & Ramdas, 2021).

In summary, the main contributions of this work are:

1. We deviate from the literature on detecting covariate or label shifts and instead focus on differentiating harmful and benign shifts. We pose the latter problem as a nonparametric sequential hypothesis test, and we differentiate between malignant and benign shifts by measuring changes in a user-specified risk metric.

2. We utilize recent progress in sequential estimation to develop tests that provably control the false alarm rate despite the multiple testing issues caused by continuously monitoring the deployed model (Section 2.2), and without constraining the form of allowed distribution shifts. For example, we do not require the target data to itself be i.i.d.; our methods are provably valid even if the target distribution is itself shifting or drifting over time.

3. We evaluate the framework on both simulated (Section 3.1) and real data (Section 3.2), illustrating its promising empirical performance. In addition to traditional losses, we also study several generalizations of the Brier score (Brier, 1950) to multiclass classification.

## 2 SEQUENTIAL TESTING FOR A SIGNIFICANT RISK INCREASE

Let $\mathcal{X}$ and $\mathcal{Y}$ denote the covariate and label spaces respectively. Consider predictors $f : \mathcal{X} \to \mathcal{Y}$. Let $\ell(\cdot, \cdot)$ be the loss function chosen to be monitored, with $R(f) := \mathbb{E}\left[\ell(f(X), Y)\right]$ denoting the corresponding expected loss, called the risk of $f$.

We assume from here onwards that $\ell$ is bounded, which is the only restriction made. This is needed to quantify how far the empirical target risk is from the true target risk without making assumptions on $P(X, Y)$. This is not a restriction of the current paper only, but appears broadly in statistical learning theory (or the study of concentration inequalities): if one picks an unbounded loss function—say the logarithmic loss—which can take on infinite values, then it is impossible to say how far the true and empirical risks are without further assumptions, for example that the distribution of $P(Y|X)$ is light tailed. We prefer not to make such assumptions in this paper, keeping it as assumption-light as possible and thus generally applicable in various domains. The restriction to bounded losses is not heavy since examples abound in the machine learning literature; see some examples in Appendix B. *Remark* 1. For classification, sometimes one does not predict a single label, but a distribution over labels. In that case the range of $f$ would be $\Delta^{|\mathcal{Y}|}$. This poses no issue, and it is common to use bounded loss functions and risks such as the Brier score, as exemplified in Section 3. On a different note, for regression, the loss (like squared error) is bounded only if the observations are. This is reasonable in some contexts (predicting rain or snow) but possibly not in others (financial losses).

### 2.1 CASTING THE DETECTION OF RISK INCREASE AS A SEQUENTIAL HYPOTHESIS TEST

We aim to trigger a warning whenever the risk on the target domain exceeds the risk on the source by a non-negligible amount specified in advance. For example, alerting could happen once it is possible to conclude with certain confidence that the accuracy has decreased by 10%. Shifts that

lead to a decrease or an insignificant increase in the risk are then treated as benign. Formally, we aim to construct a sequential test for the following pair of hypotheses:

$$H_0: \quad R_T(f) \leq R_S(f) + \varepsilon_{\text{tol}}, \quad \text{vs.} \quad H_1: \quad R_T(f) > R_S(f) + \varepsilon_{\text{tol}}, \quad (1)$$

where $\varepsilon_{\text{tol}} \geq 0$ is an acceptable tolerance level, and $R_S(f)$ and $R_T(f)$ stand for the risk of $f$ on the source and target domains respectively. Assume that one observes a sequence of data points $Z_1, Z_2, \ldots$. At each time point $t$, a sequential test takes the first $t$ elements of this sequence and output either a 0 (continue) or 1 (reject the null and stop). The resulting sequence of 0s and 1s satisfies the property that if the null $H_0$ is true, then the probability that the test ever outputs a 1 and stops (false alarm) is at most $\delta$. In our context, this means that if a distribution shift is benign, then with high probability, the test will never output a 1 and stop, and thus runs forever. Formally, a level-$\delta$ sequential test $\Phi$ defined as a mapping $\bigcup_{n=1}^{\infty} \mathcal{Z}^n \to \{0, 1\}$ must satisfy: $\mathbb{P}_{H_0}(\exists t \geq 1 : \Phi(Z_1, \ldots, Z_t) = 1) \leq \delta$. Note that the sequential nature of a test is critical here as we aim to develop a framework capable of continuously updating inference as data from the target is collected, making it suitable for many practical scenarios. We distinguish between two important settings when we assume that the target data either satisfies (a) an i.i.d. assumption (under the same or different distribution as the source) or (b) only an independence assumption. While the i.i.d. assumption may be arguably reasonable on the source, it is usually less realistic on the target, since in practice, one may expect the distribution to drift slowly in a non-i.i.d. fashion instead of shifting sharply but staying i.i.d.. Under setting (b), the quantity of interest on the target domain is the *running risk*:

$$R^{(t)}(f) = \frac{1}{t} \sum_{i=1}^{t} \mathbb{E}\left[\ell\left(f(X_i'), Y_i'\right)\right], \quad t \geq 1,$$

where the expected value is taken with respect to the joint distribution of $(X_i', Y_i')$, possibly different for each test point $i$. The goal transforms into designing a test for the following pair of hypotheses:

$$H_0: \quad R_T^{(t)}(f) \leq R_S(f) + \varepsilon_{\text{tol}}, \ \forall t \geq 1, \quad \text{vs.} \quad H_1: \quad \exists t^\star \geq 1 : R_T^{(t^\star)}(f) > R_S(f) + \varepsilon_{\text{tol}}. \quad (2)$$

When considering other notions of risk beyond the misclassification error, one could also be interested in relative changes in the risk, and thus a sequential test for the following pair of hypotheses:

$$H_0': \quad R_T(f) \leq (1 + \varepsilon_{\text{tol}})R_S(f), \quad \text{vs.} \quad H_1': \quad R_T(f) > (1 + \varepsilon_{\text{tol}})R_S(f). \quad (3)$$

The proposed framework handles all of the aforementioned settings as we discuss next. The most classical approach for sequential testing is the sequential probability ratio test (SPRT) due to Wald (1945). However, it can only be applied, even for a point null and a point alternative, when the relevant underlying distributions are known. While extensions of the SPRT exist to the composite null and alternative (our setting above), these also require knowledge of the distributions of the test statistics (e.g., empirical risk) under the null and alternative, and being able to maximize the likelihood. Clearly, we make no distributional assumptions and so we require a nonparametric approach. We perform sequential testing via the dual problem of nonparametric sequential estimation, a problem for which there has been much recent progress to draw from.

## 2.2 Sequential testing via sequential estimation

When addressing a particular prediction problem, the true risk on neither the source nor the target domains is known. Performance of a model on the source domain is usually assessed through a labeled holdout source sample of a fixed size $n_S$: $\{(X_i, Y_i)\}_{i=1}^{n_S}$. We can write:

$$R_S(f) + \varepsilon_{\text{tol}} = \widehat{R}_S(f) + \left(R_S(f) - \widehat{R}_S(f)\right) + \varepsilon_{\text{tol}},$$

where $\widehat{R}_S(f) := \left(\sum_{i=1}^{n_S} \ell(f(X_i), Y_i)\right)/n_S$. For any fixed tolerance level $\delta_S \in (0, 1)$, classic concentration results can be used to obtain an upper confidence bound $\varepsilon_{\text{appr}}$ on the difference $R_S(f) - \widehat{R}_S(f)$, and thus to conclude that with probability at least $1 - \delta_S$:

$$R_S(f) + \varepsilon_{\text{tol}} \leq \widehat{U}_S(f) + \varepsilon_{\text{tol}}, \quad \text{where} \quad \widehat{U}_S(f) = \widehat{R}_S(f) + \varepsilon_{\text{appr}}. \quad (4)$$

For example, by Hoeffding's inequality, $n_S = O(1/\varepsilon_{\text{appr}}^2)$ points suffice for the above guarantee, but that bound can be quite loose when the individual losses $\ell(f(X_i), Y_i)$ have low variance. In such settings, recent variance-adaptive confidence bounds (Waudby-Smith & Ramdas, 2021; Howard et al.,

2021) are tighter. It translates to an increase in the power of the framework, allowing for detecting harmful shifts much earlier, while still controlling the false alarm rate at a prespecified level.

In contrast, the estimator of the target risk has to be updated as losses on test instances are observed. While the classic concentration results require specifying in advance the size of a sample used for estimation, time-uniform confidence sequences retain validity under adaptive data collection settings. For any chosen $\delta_T \in (0,1)$, those yield a time-uniform lower confidence bound on $R_T(f)$:

$$\mathbb{P}\left(\exists t \geq 1 : R_T(f) < \widehat{L}_T^{(t)}(f)\right) \leq \delta_T,$$

where $\widehat{L}_T^{(t)}(f)$ is the bound constructed after processing $t$ test points. We typically set $\delta_S = \delta_T = \delta/2$, where $\delta$ refers to the desired type I error. Under the independence assumption (equation 2), the form of the drift in the distribution of the data is allowed to change with time. From the technical perspective, the difference with the i.i.d. setting is given by the applicability of particular concentration results. While the betting-based approach of Waudby-Smith & Ramdas (2021) necessitates assuming that random variables share a common mean, proceeding with the conjugate-mixture empirical-Bernstein bounds (Howard et al., 2021) allows us to lift the common-mean assumption and handle a time-varying mean. We summarize the testing protocol in Algorithm 1 and refer the reader to Appendix E for a review of the concentration results used in this work. One can easily adapt the framework to proceed with a fixed, absolute threshold on the risk, rather than a relative threshold $R_S(f) + \varepsilon_{\text{tol}}$, e.g., we can raise a warning once accuracy drops below 80%, rather than 5% below the training accuracy.

---

**Algorithm 1** Sequential testing for an absolute increase in the risk.

    **Input:** Predictor $f$, loss $\ell$, tolerance level $\varepsilon_{\text{tol}}$, sample from the source $\{(X_i, Y_i)\}_{i=1}^{n_S}$.
1: **procedure**
2:     Compute the upper confidence bound on the source risk $\widehat{U}_S(f)$;
3:     **for** $t = 1, 2, \ldots$ **do**
4:         Compute the lower confidence bound on the target risk $\widehat{L}_T^{(t)}(f)$;
5:         **if** $\widehat{L}_T^{(t)}(f) > \widehat{U}_S(f) + \varepsilon_{\text{tol}}$ **then**
6:             Reject $H_0$ (equation 1) and fire off a warning.

---

Testing for a relative increase in the risk is performed by replacing the line 5 in the Algorithm 1 by the condition $\widehat{L}_T^{(t)}(f) > (1 + \varepsilon_{\text{tol}})\widehat{U}_S(f)$. For both cases, the proposed test controls type I error as formally stated next. The proof is presented in Appendix D.

**Proposition 1.** *Fix any $\delta \in (0,1)$. Let $\delta_S, \delta_T \in (0,1)$ be chosen in a way such that $\delta_S + \delta_T = \delta$. Let $\widehat{L}_T^{(t)}(f)$ define a time-uniform lower confidence bound on $R_T^{(t)}(f)$ at level $\delta_T$ after processing $t$ data points ($t \geq 1$), and let $\widehat{U}_S(f)$ define an upper confidence bound on $R_S(f)$ at level $\delta_S$. Then:*

$$\begin{cases} \mathbb{P}_{H_0}\left(\exists t \geq 1 : \widehat{L}_T^{(t)}(f) > \widehat{U}_S(f) + \varepsilon_{tol}\right) \leq \delta, \\ \mathbb{P}_{H_0'}\left(\exists t \geq 1 : \widehat{L}_T^{(t)}(f) > (1 + \varepsilon_{tol})\widehat{U}_S(f)\right) \leq \delta, \end{cases} \tag{5}$$

*that is, the procedure described in Algorithm 1 controls the type I error for testing the hypotheses $H_0$ (equation 1 and equation 2) and $H_0'$ (equation 3).*

*Remark* 2. Both the testing protocol (Algorithm 1) and the corresponding guarantee (Proposition 1) are stated in a form which requires the lower bound on the target risk to be recomputed after processing each test point. More generally, test data could be processed in minibatches of size $m \geq 1$.

*Remark* 3. Type I error guarantee in equation 5 holds under continuous monitoring. This goes beyond standard fixed-time guarantees, for which type I error is controlled only when the sample size is fixed in advance, and not under continuous monitoring. Define a stopping time of a sequential test in Algorithm 1:

$$N(\delta) := \inf\left\{t \geq 1 : \widehat{L}_T^{(t)}(f) > \widehat{U}_S(f) + \varepsilon_{\text{tol}}\right\}.$$

Then the guarantee in equation 5 can be restated as: $\mathbb{P}_{H_0}(N(\delta) < \infty) \leq \delta$, that is the probability of ever raising a false alarm is at most $\delta$.

**From sequential testing to changepoint detection.** A valid sequential test can be transformed into a changepoint detection procedure with certain guarantees (Lorden, 1971). The key characteristics of changepoint detection procedures are *average run length* (ARL), or average time to a false alarm, and *average detection delay* (ADD). One way to convert a sequential test into a detection procedure is by running a separate test starting at each time point $t = 1, 2, \ldots$, and claiming a change whenever the first one of the tests rejects the null. Subsequently, these tests yield a sequence of stopping variables $N_1(\delta), N_2(\delta), \ldots$ The corresponding stopping time is defined as:

$$N^\star(\delta) := \inf_{k=1,2,\ldots} \left( N_k(\delta) + (k-1) \right).$$

Lorden (1971) established a lower bound on the (worst-case) ARL of such changepoint detection procedure of the form: $\mathbb{E}_{H_0}\left[ N^\star(\delta) \right] \geq 1/\delta$. The (worst-case) average detection delay is defined as:

$$\overline{\mathbb{E}}_1 N(\delta) = \sup_{m \geq 1} \operatorname{ess\,sup} \mathbb{E}_m \left[ (N(\delta) - (m-1))_+ \mid (X_1', Y_1'), \ldots, (X_{m-1}', Y_{m-1}') \right],$$

where $\mathbb{E}_m$ denotes expectation under $P_m$, the distribution of a sequence $(X_1', Y_1'), (X_1', Y_1'), \ldots$ under which $(X_m', Y_m')$ is the first term from a shifted distribution.

## 3 EXPERIMENTS

In Section 3.1, we analyze the performance of the testing procedure on a collection of simulated datasets. First, we consider settings where the i.i.d. assumption on the target holds, and then relax it to the independence assumption. In Section 3.2, we evaluate the framework on real data. We consider classification problems with different metrics of interest including misclassification loss, several versions of the Brier score and miscoverage loss for set-valued predictors. Due to space limitations, we refer the reader to Appendix B for a detailed review of the considered loss functions.

### 3.1 SIMULATED DATA

**Tracking the risk under the i.i.d. assumption.** Here we induce label shift on the target domain and emulate a setting where it noticeably harms the accuracy of a predictor by modifying the setup from Section 1 through updating the class centers to $\mu_0 = (-1, 0)^\top$ and $\mu_1 = (1, 0)^\top$, making the classes largely overlap. The (oracle) Bayes-optimal predictor on the source domain is:

$$f^\star(x) = \frac{\pi_1^S \cdot \varphi(x; \mu_1, I_2)}{\pi_0^S \cdot \varphi(x; \mu_0, I_2) + \pi_1^S \cdot \varphi(x; \mu_1, I_2)}, \tag{6}$$

where $\varphi(x; \mu_i, I_2)$ denotes the probability density function of a Gaussian random vector with mean $\mu_i$, $i \in \{0, 1\}$ and an identity covariate matrix. Let $\ell$ be the 0-1 loss, and thus the misclassification risk $R_T(f^*)$ on the target is:

$$
\mathbb{P}_T \left( f^\star(X) \neq Y \right) = \mathbb{P}\left( X^\top (\mu_1 - \mu_0) < \log\left( \frac{\pi_0^S}{\pi_1^S} \right) + \frac{1}{2} \left( \|\mu_1\|_2^2 - \|\mu_0\|_2^2 \right) \mid Y = 1 \right) \cdot \pi_1^T
$$
$$
+ \mathbb{P}\left( X^\top (\mu_1 - \mu_0) \geq \log\left( \frac{\pi_0^S}{\pi_1^S} \right) + \frac{1}{2} \left( \|\mu_1\|_2^2 - \|\mu_0\|_2^2 \right) \mid Y = 0 \right) \cdot \pi_0^T.
$$

For three values of $\pi_1^S$, the source marginal probability of class 1, we illustrate how label shift affects the misclassification risk of the Bayes-optimal predictor on Figure 2a, noting that it is linear in $\pi_1^T$. Importantly, whether label shift hurts or helps depends on the value of $\pi_1^S$.

We fix $\pi_1^S = 0.25$ and use the corresponding Bayes-optimal rule. On Figure 2b, we compare upper confidence bounds on the source risk due to different concentration results, against the size of the source holdout set. Variance-adaptive upper confidence bounds—predictably-mixed empirical-Bernstein (PM-EB) and betting-based (see Appendix E)—are much tighter than the non-adaptive Hoeffding's bound. Going forward, we use a source holdout set of 1000 points to compute upper confidence bound on the source risk, where $\varepsilon_{\mathrm{appr}}$ from equation 4 is around 0.025.

Next, we fix $\varepsilon_{\mathrm{tol}} = 0.05$, i.e., we treat a 5% drop in accuracy as significant. For 20 values of $\pi_1^T$, evenly spaced in the interval $[0.1, 0.9]$, we sample 40 batches of 50 data points from the target

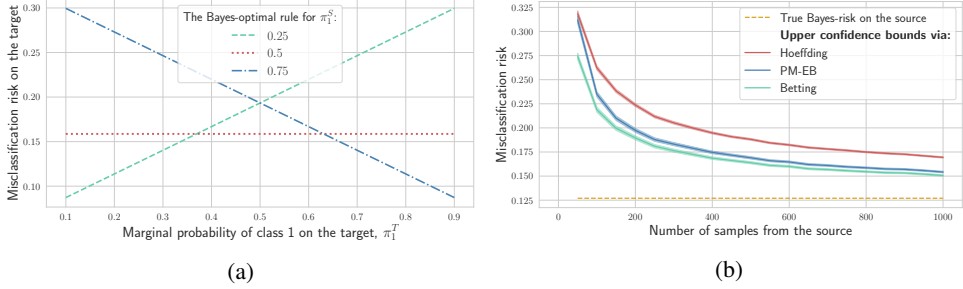

(a)                                                                (b)

Figure 2: (a) The misclassification risk on the target of the Bayes-optimal predictors for three values of $\pi_1^S$. Notice that label shift does not necessarily lead to an increase in the risk. (b) Upper confidence bounds $\widehat{U}_S(f)$ on the misclassification risk on the source obtained via several possible concentration results. For each sample size, the results are aggregated over 1000 random data draws. The variance-adaptive confidence bounds (predictably-mixed empirical-Bernstein and the betting-based one) are much tighter when compared against the non-adaptive one.

distribution. On Figure 3a, we track the proportion of null rejections after repeating the process 250 times. Note that here stronger label shift hurts accuracy more. On Figure 3b, we illustrate average size of a sample from the target needed to reject the null. The results confirm that tighter bounds yield better detection procedures, with the most powerful test utilizing the betting-based bounds (Waudby-Smith & Ramdas, 2021). A similar analysis for the Brier score (Brier, 1950) as a target metric is presented in Appendix F. We further study the performance of the framework under the covariate shift setting (Appendix H).

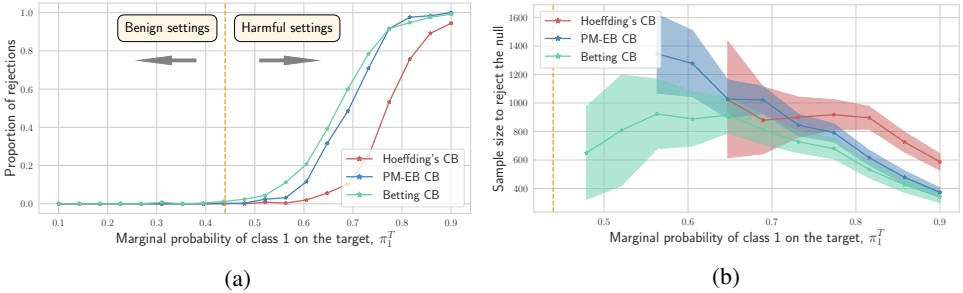

(a)                                                                (b)

Figure 3: (a) Proportion of null rejections when testing for an increase in the misclassification risk after processing 2000 samples from a shifted distribution. The vertical dashed yellow line separates null (benign) and alternative (harmful) settings. Testing procedures that rely on variance-adaptive confidence bounds (CBs) have more power. (b) Average sample size from the target that was needed to reject the null. Tighter concentration results allow to raise an alarm after processing less samples.

**Tracking beyond the i.i.d. setting (distribution *drift*).** Here we consider testing for an increase in the running risk (equation 2). First, we fix $\pi_1^T = 0.75$ and keep the data generation pipeline as before, that is, the target data are still sampled in an i.i.d. fashion. We compare lower confidence bounds on the target risk studied before against the conjugate-mixture empirical-Bernstein (CM-EB) bound (Howard et al., 2021). We note that this bound on the running mean of the random variables does not have a closed-form and has to be computed numerically. On Figure 4a, we illustrate that the lower confidence bounds based on betting are generally tighter only for a small number of samples. Similar results hold for the Brier score as a target metric (see Appendix F.1).

Next, we lift the i.i.d. assumption by modifying the data generation pipeline: starting with $\pi_1^T = 0.25$, we increase $\pi_1^T$ by 0.1 after sampling each 200 instances, until it reaches the value 0.85. It makes CM-EB the only valid lower bound on the running risk on the target domain. The results of running the framework for this setting are presented on Figure 4b.

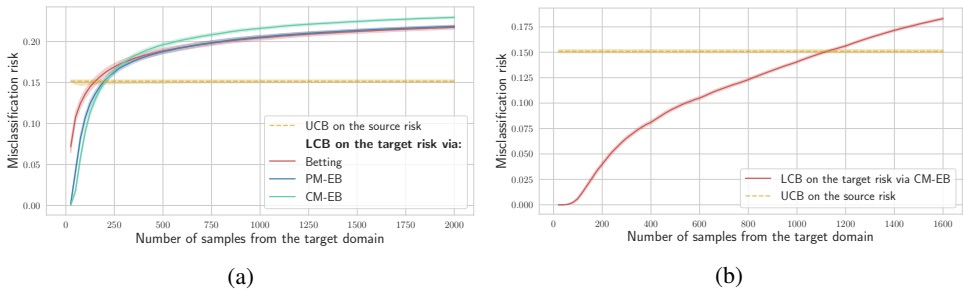

(a)                 (b)

Figure 4: (a) Different lower confidence bounds (LCB) on the target risk under the i.i.d. assumption. Betting-based LCB is only tighter than conjugate-mixture empirical-Bernstein (CM-EB) for a small number of samples. (b) Under distribution drift, only CM-EB performs estimation of the running risk. The resulting test consistently detects a harmful increase in the running risk.

## 3.2 REAL DATA

In deep learning, out-of-distribution robustness is often assessed based on a model performance gap between the original data (used for training) and data to which various perturbations are applied. We focus on two image classification datasets with induced corruptions: MNIST-C (Mu & Gilmer, 2019) and CIFAR-10-C (Krizhevsky, 2009; Hendrycks & Dietterich, 2019). We illustrate an example of a clean MNIST image on Figure 5a along with its corrupted versions after applying *motion blur*, blur along a random direction (Figure 5b), *translate*, affine transformation along a random direction (Figure 5c), and *zigzag*, randomly oriented zigzag over an image (Figure 5d). For CIFAR-10-C, we consider corrupting original images by applying the *fog* effect with 3 levels of severity as illustrated on the bottom row of Figure 5. For both cases, clean or corrupted test samples are passed as input to networks trained on clean data. While corruptions are visible to the human eye, one might still hope that they will not significantly hurt classification performance. We use the betting-based confidence bounds on the source and conjugate-mixture empirical-Bernstein confidence bounds on the target domain.

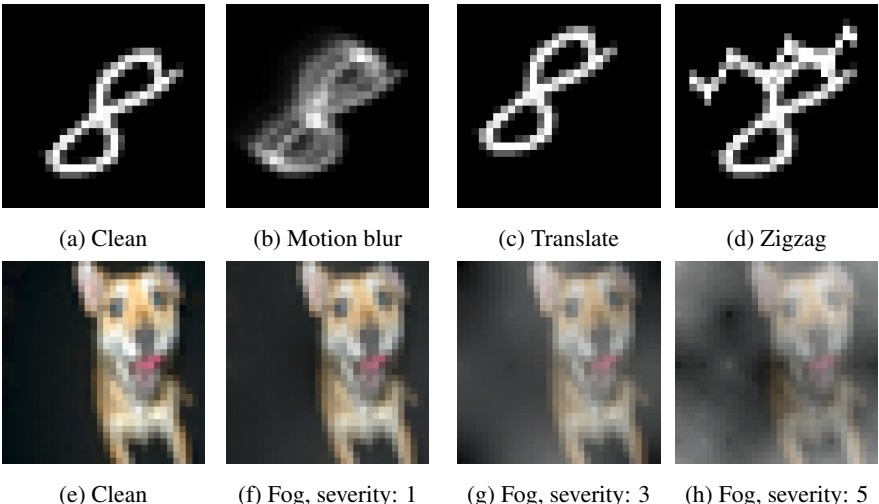

(a) Clean       (b) Motion blur       (c) Translate       (d) Zigzag

(e) Clean       (f) Fog, severity: 1       (g) Fog, severity: 3       (h) Fog, severity: 5

Figure 5: Examples of MNIST-C ((a)–(d)) and CIFAR-10-C ((e)–(h)) images.

**Tracking the risk of a point predictor on MNIST-C dataset.** We train a shallow CNN on clean MNIST data and run the framework testing whether the misclassification risk increases by 10%, feeding the network with data in batches of 50 points either from the original or shifted distributions. Details regarding the network architecture and the training process are given in Appendix G. On Figure 6a, we illustrate the results after running the procedure 50 times for each of the settings.

The horizontal dashed line defines the rejection threshold that has been computed using the source data: once the lower bound on the target risk (solid lines) exceeds this value, the null hypothesis is rejected. When passing clean MNIST data as input, we do not observe degrading performance. Further, applying different corruptions leads to both benign and harmful shifts. While to the human eye the translate effect is arguably the least harmful one, it is the most harmful to the performance of a network. Such observation is also consistent with findings of Mu & Gilmer (2019) who observe that if trained on clean MNIST data without any data augmentation, CNNs tend to fail under this corruption. We validate this observation by retraining the network several times in Appendix G.

**Tracking the risk of a set-valued predictor on CIFAR-10-C dataset.**  For high-consequence settings, building accurate models only can be insufficient as it is crucial to quantify uncertainty in predictions. One way to proceed is to output a set of candidate labels for each point as a prediction. The goal could be to cover the correct label of a test point with high probability (Vovk et al., 2005) or control other notions of risk (Bates et al., 2021). We follow Bates et al. (2021) who design a procedure that uses a holdout set for tuning the parameters of a wrapper built on top of the original model which, under the i.i.d. assumption, is guaranteed to have low risk with high probability (see Appendix G for details). Below, we build a wrapper around a ResNet-32 model that controls the miscoverage risk (equation 14) at level $0.1$ with probability at least $0.95$. For each run, CIFAR-10 test set is split at random into three folds used for: (a) learning a wrapper (1000 points), (b) estimating upper confidence bound on the miscoverage risk on the source (1000 points), and (c) evaluation purposes on either clean or corrupted images. We take $\varepsilon_{\text{tol}} = 0.05$, that is, 5% drop in coverage is treated as significant. Figure 6b illustrates that only the most intense level of fog is consistently harmful to coverage. We also consider setting a lower prescribed miscoverage level (0.05) for the set-valued predictor (see Appendix G). When larger prediction sets are produced, adding fog to images becomes less harmful.

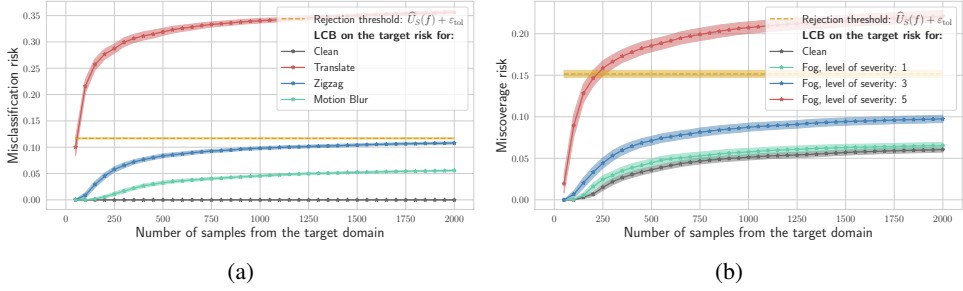

(a)                                    (b)

Figure 6: (a) Performance of the testing framework on MNIST-C dataset. Only the translation effect is consistently harmful to the classification performance of a CNN trained on clean data. (b) Performance of the testing framework on CIFAR-10-C dataset. Only the most severe version of the fog lead to a significant degradation in performance measured by a decrease in coverage of a set-valued predictor trained on top of a model trained on clean data.

## 4  CONCLUSION

An important component of building reliable machine learning systems is making them alarm a user when potentially unsafe behavior is observed, instead of allowing them to fail silently. Ideally, a warning should be displayed when critical changes affecting model performance are present, e.g., a significant degradation of the target performance metrics, like accuracy or calibration. In this work, we considered one particular failure scenario of deployed models—presence of distribution shifts. Relying solely on point estimators of the performance metrics ignores uncertainty in the evaluation, and thus fails to represent a theoretically grounded approach. We developed a set of tools for deciding whether the performance of a model on the test data becomes significantly worse than the performance on the training data in a data-adaptive way. The proposed framework based on performing sequential estimation requires observing true labels for test data (possibly, in a delayed fashion). Across various types of distribution shifts considered in this work, it demonstrated promising empirical performance for differentiating between harmful and benign ones.

**Acknowledgements**   The authors thank Ian Waudby-Smith for fruitful discussions and the anonymous ICLR 2022 reviewers for comments on an early version of this paper. The authors acknowledge support from NSF DMS 1916320 and an ARL IoBT CRA grant. Research reported in this paper was sponsored in part by the DEVCOM Army Research Laboratory under Cooperative Agreement W911NF-17-2-0196 (ARL IoBT CRA). The views and conclusions contained in this document are those of the authors and should not be interpreted as representing the official policies, either expressed or implied, of the Army Research Laboratory or the U.S. Government. The U.S. Government is authorized to reproduce and distribute reprints for Government purposes notwithstanding any copyright notation herein.

**Ethics Statement.**   We do not foresee any negative impacts of this work in and of itself; there may be societal impacts of the underlying classification or regression task but these are extraneous to the aims and contributions of this paper.

**Reproducibility statement.**   In order to ensure reproducibility of the results in this paper, we include the following to the supplementary materials: (a) relevant source code for all simulations that have been performed, (b) clear explanations of all data generation pipelines and preprocessing steps that have been performed, (c) complete proofs for all established theoretical results.

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

## A  ISSUES WITH EXISTING TESTS FOR DISTRIBUTION SHIFTS/DRIFTS

### A.1  NON-SEQUENTIAL TESTS HAVE HIGHLY INFLATED FALSE ALARM RATES WHEN DEPLOYED IN SEQUENTIAL SETTINGS WITH CONTINUOUS MONITORING

In this work, we propose a framework that utilizes confidence sequences (CSs), and thus allows for continuous monitoring of model performance. On the other hand, traditional (fixed-time) testing procedures are not valid under sequential settings, unless corrections for multiple testing are performed. First, we illustrate that deploying fixed-time detection procedures under sequential settings necessarily leads to raising false alarms. Then, we illustrate that naive corrections for multiple testing—without taking advantage of the dependence between the tests—lead to losses of power of the resulting procedure.

**Deploying fixed-time tests without corrections for multiple testing.**  Under the i.i.d. setting, our framework reduces to testing whether the means corresponding to two unknown distributions are significantly different. Here we consider a simplified setting: assume that one observes a sequence $Z_1, Z_2, \ldots$ of bounded i.i.d. random variables, and the goal is to construct a lower confidence bound for the corresponding mean $\mu$. In this case, a natural alternative to confidence sequences is a lower bound obtained by invoking the Central Limit Theorem:

$$\overline{Z}_t - z_\delta \cdot \frac{\widehat{\sigma}_t}{\sqrt{t}},$$

where $z_\delta$ is $(1 - \delta)$-quantile of the standard Gaussian random variable and $\overline{Z}_t, \widehat{\sigma}_t$ denote the sample average and sample standard deviation respectively computed using first $t$ instances $Z_1, \ldots, Z_t$. For the study below, we use the critical (significance) level $\delta = 0.1$. We sample the data as: $Z_t \sim$ Ber$(0.6)$, $t = 1, 2, \ldots$, and consider 100 possible sample sizes, evenly spaced between 20 and 1000 on a logarithmic scale. Next, we compare the CLT lower bound with the betting-based one (which underlies the framework proposed in this work) under the following settings:

1. **Fixed-time monitoring.** For a given sample size $t$, we sample the sequence $Z_1, \ldots, Z_t$, compute the lower bounds and check whether the true mean is covered only once. For each value of the sample size, we resample data 100 times and record the miscoverage rate, that is, the fraction of times the true mean is miscovered.

2. **Continuous monitoring.** Here, the lower bound is recomputed once new data become available. We resample the whole data sequence $Z_1, \ldots, Z_{1000}$ 1000 times, and for each value of the sample size, we track the *cumulative* miscoverage rate, that is, the fraction of times the true mean has been miscovered *at some time* up to $t$.

Under fixed-time monitoring (Figure 7a), the false alarm rate is controlled at prespecified level $\delta$ by both procedures. However, under continuous monitoring (Figure 7b), deploying the CLT lower bound leads to raising false alarms. At the same time, the betting-based lower bound controls the false alarm rate under both types of monitoring.

**Deploying fixed-time tests with corrections for multiple testing.**  Next, we illustrate that adapting fixed-time tests to sequential settings via performing corrections for multiple testing comes at the price of significant power losses. Performing the Bonferroni correction requires splitting the available error budget $\delta$ among the times when testing is performed. In particular, we consider:

$$\textbf{Power correction:} \quad \sum_{i=1}^{\infty} \frac{\delta}{2^i} = \delta,$$

$$\textbf{Polynomial correction:} \quad \frac{6}{\pi^2} \sum_{i=1}^{\infty} \frac{\delta}{i^2} = \delta. \tag{7}$$

Note that the second option is preferable as the terms in the sum decrease at a slower rate, thus allowing for a narrower sequence of intervals. Proceeding under the setup considered in the beginning of this section (data points are sampled from Ber$(0.6)$), we consider two scenarios:

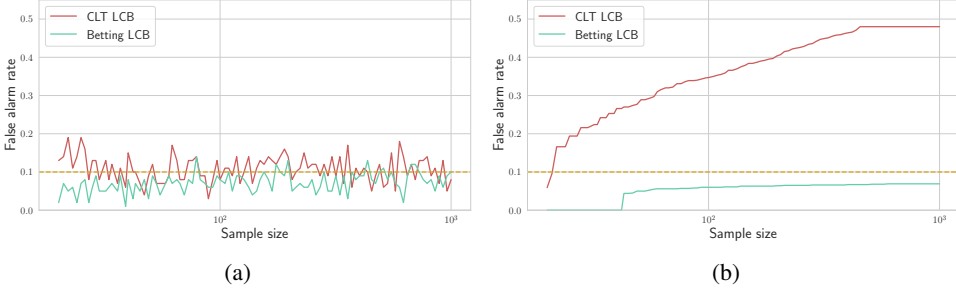

(a)                                               (b)

Figure 7: False alarm rate for the CLT and betting-based lower confidence bound (LCB) under: (a) fixed-time monitoring and (b) continuous monitoring. Note that both bounds control the false alarm rate at a prespecified level $\delta = 0.1$ under fixed-time monitoring. However under continuous monitoring, the false alarm rate of the CLT bound quickly exceeds the critical level $\delta = 0.1$. At the same time, the betting LCB successfully controls the false alarm rate.

1. We recompute the CLT lower bound each time a batch of 25 samples is received and perform the Bonferroni correction (utilizing both ways of splitting the error budget described in equation 7). We present the lower bounds on Figure 8a (the results have been aggregated over 100 data draws). Observe that:

   - While the sequence of intervals shrinks in size with growing number of samples under the polynomial correction, this is not the case under the power correction.
   - Not only the betting confidence sequence is uniformly tighter than the CLT-based over all considered sample sizes, it also allows for monitoring at arbitrary stopping times. Note that the CLT lower bound allows for monitoring only at certain times (marked with stars on Figure 8a).

2. For a fairer comparison with the betting-based bound, we also consider recomputing the CLT bound each time a new sample is received. Since utilizing the power correction quickly leads to numerical overflows, we utilize only the polynomial correction. We present the lower bounds on Figure 8b (the results have been aggregated over 100 data draws). While now the CLT lower bound can be monitored at arbitrary stopping times, it is substantially lower (thus more conservative) than the betting-based.

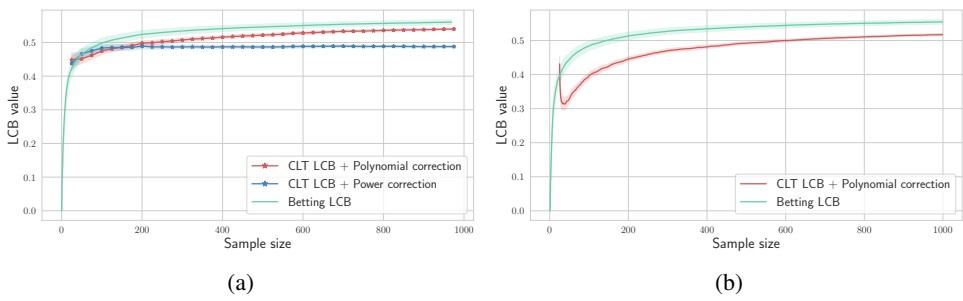

(a)                                               (b)

Figure 8: Adapting the CLT lower bound to continuous monitoring via performing corrections for multiple testing: (a) each time a batch of 25 samples is received, (b) each time a new sample is received. Under both settings, the CLT-based lower bound is more conservative than the betting-based, which, in testing terminology, means that the resulting testing framework has less power.

## A.2  CONFORMAL TEST MARTINGALES DO NOT IN GENERAL DIFFERENTIATE BETWEEN HARMFUL AND BENIGN DISTRIBUTION SHIFTS

Testing the i.i.d. assumption online can be performed using conformal test martingales (Vovk et al., 2021; Vovk, 2021; 2020). Below, we review building blocks underlying a conformal test martingale.

1. First, one has to pick a *conformity score*. Assigning a lower score to a sample indicates abnormal behavior. Vovk et al. (2021) consider the regression setting and, like us, use scores that depends on true labels: for a point $(x_i, y_i)$, let $\widehat{y}_i$ denote the output of a predictor on input $x_i$. The authors propose a score of the form:

$$\alpha_i = -|y_i - \widehat{y}_i|. \tag{8}$$

   Note that lower scores defined in equation 8 clearly reflect degrading performance of a predictor (possibly due to the presence of distribution drift). Under the classification setting, we propose to consider the following conformity score:

$$\alpha_i = \sum_{i=1}^{K} f_k(x_i) \cdot \mathbb{1}\{f_k(x_i) \leq f_{y_i}(x_i)\} = 1 - \sum_{i=1}^{K} f_k(x_i) \cdot \mathbb{1}\{f_k(x_i) > f_{y_i}(x_i)\}, \tag{9}$$

   which is a rescaled estimated probability mass of all the labels that are more likely than the true one (here, we assume that predictor $f$ outputs an element of $\Delta^{|\mathcal{Y}|}$). Rescaling in equation 9 is used to ensure that this score represent the *conformity* score, that is, the higher the value is, the better a given data point conforms. Note that if for a given data point, the true label happens to be top-ranked by a predictor $f$, then such point receives the largest conformity score equal to one. This score is inspired by recent works in conformal classification (Romano et al., 2020; Podkopaev & Ramdas, 2021).

2. After processing $n$ data points, a *transducer* transforms a collection of conformity scores into a conformal p-value:

$$P_n = p\left(\{(x_i, y_i)\}_{i=1}^n, u\right) := \frac{|i \in \{1, \ldots, n\} : \alpha_i < \alpha_n| + u \cdot |i \in \{1, \ldots, n\} : \alpha_i = \alpha_n|}{n},$$

   where $u \sim \text{Unif}([0, 1])$. Conformal p-values are i.i.d. uniform on $[0, 1]$ when the data points are i.i.d. (or more generally, exchangeable; see (Vovk et al., 2021)). Note that the design of conformal p-value $P_n$ ensures it takes small value when the conformity score $\alpha_n$ is small, that is, when abnormal behavior is being observed in a sequence.

3. A betting martingale is used to gamble again the null hypothesis that a sequence of random variables is distributed uniformly and independently on $[0, 1]$. Formally, a betting martingale is a measurable function $F : [0, 1]^\star \to [0, \infty]$ such that $F(\square) = 1$ ($\square$ defines an empty sequence and $Z^\star$ stands for the set of all finite sequences of elements of $Z$) and for each sequence $(u_1, \ldots, u_{n-1}) \in [0, 1]^{n-1}$ and any $n \geq 1$:

$$\int_0^1 F(u_1, \ldots, u_{n-1}, u) du = F(u_1, \ldots, u_{n-1}).$$

   The simplest example is given by the product of simple bets:

$$F(u_1, \ldots, u_n) = \varepsilon^n \left(\prod_{i=1}^n u_i\right)^{1-\varepsilon}, \quad \varepsilon > 0, \tag{10}$$

   but more sophisticated options are available (Vovk et al., 2005). For the simulations that follow, we use simple mixture martingale which is obtained by integrating equation 10 over $\varepsilon \in [0, 1]$.

   Conformal test martingale $S_n$ is obtained by plugging in the sequence of conformal p-values $P_1, \ldots, P_n$ into the betting martingale. The test starts with $S_0 = 1$ and it rejects at the first time $n$ when $S_n$ exceeds $1/\alpha$. They type I error control for this test is justified by Ville's inequality which states that for any nonnegative martingale (which $S_n$ is, under the i.i.d. null), the entire process $S_n$ stays below $1/\alpha$ with probability at least $1 - \alpha$. Mathematically:

$$\mathbb{P}(\exists n : S_n \geq 1/\alpha) \leq \alpha.$$

To study conformal test martingales, we consider the label shift setting described in Section 3.1. Recall that for this setting we know exactly when a shift in label proportions becomes harmful to misclassification risk of the Bayes-optimal rule on the source distribution (see Figures 2a and 3a). For the simulations that follow, we assume that the marginal probability of class 1 on the source is $\pi_1^S = 0.25$, and use the corresponding optimal rule.

We analyze conformal test martingales under several distribution drift scenarios differing in their severity and rate, and start with the settings where a sharp shift is present.

1. ***Harmful* distribution *shift* with *cold start*.** Here, the data are sampled i.i.d. from a shifted distribution corresponding to $\pi_1^T = 0.75$. We illustrate 50 runs of the procedure on Figure 9a. Recall that when the data are sampled i.i.d. the conformal p-values are i.i.d. uniform on $[0, 1]$. Under the (exchangeability) null, conformal test martingales are not growing, and thus are not able to detect that a present shift, even though it corresponds to a harmful setting.

2. ***Harmful* distribution *shift* with *warm start*.** For a fairer comparison, we also consider a warm start setting when the first 100 points are sampled i.i.d. from the source distribution ($\pi_1^T = 0.25$), followed by the data sampled i.i.d. from a shifted distribution ($\pi_1^T = 0.75$). We illustrate 50 runs of the procedure on Figure 9b. In this case, conformal test martingales demonstrate better detection properties. However, a large fraction of conformal test martingales still is incapable of detecting a shift.

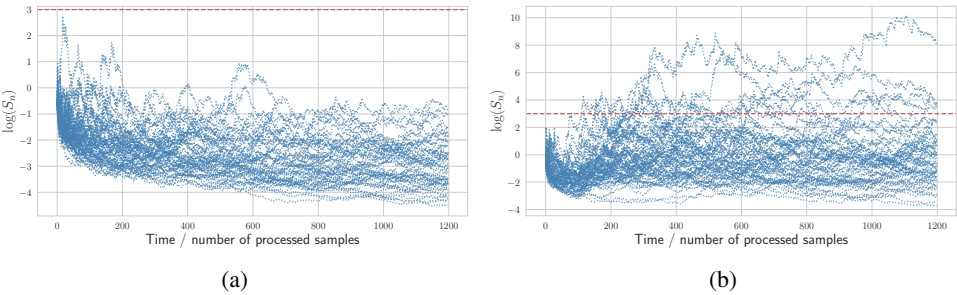

(a)                                        (b)

Figure 9: 50 runs of conformal test martingales (blue dotted lines) under harmful distribution shift with: (a) cold start (shift happens in the beginning), (b) warm start (shift happens in an early stage of a model deployment). The horizontal red dashed line outlines to the rejection threshold due to Ville's inequality. Even though warm start improves detection properties, only a small fraction of conformal test martingales detects a shift that leads to more than 10% drop in classification accuracy.

The simulations above illustrate that conformal test martingales are inferior to the framework proposed in this work whenever a sharp distribution shift happens in the early stage of a model deployment, even when such shift is harmful. Next, we consider several settings where instead of changing sharply, the distribution drifts gradually.

3. ***Slow* and *benign* distribution *drift*.** Starting with the marginal probability of class 1, $\pi_1^T = 0.1$, we keep *increasing* $\pi_1^T$ by 0.05 each time a batch of 75 data points is sampled until it reaches the value 0.45. Recall from Section 3.1 that those values of $\pi_1^T$ correspond to a *benign* setting where the risk of the predictor on the target domain does not exceed substantially the source risk. We illustrate 50 runs of the procedure on Figure 10a.

4. ***Slow* and *harmful* distribution *drift*.** Starting with the marginal probability of class 1 $\pi_1^T = 0.5$, we keep *increasing* $\pi_1^T$ by 0.05 each time a batch of 75 data points is sampled until it reaches the value 0.85. Recall from Section 3.1 that those values of $\pi_1^T$ correspond to a *harmful* setting where the risk of the predictor on the target domain is substantially larger than the source risk. We illustrate 50 runs of the procedure on Figure 10b.

5. ***Sharp* and *harmful* distribution *drift*.** Starting with the marginal probability of class 1, $\pi_1^T = 0.1$, we keep *increasing* $\pi_1^T$ by 0.2 each time a batch of 150 data points is sampled until it reaches the value 0.9. We illustrate 50 runs of the procedure on Figure 10c.

The settings where the distribution drifts gradually illustrate several shortcomings of conformal test martingales:

- Conformal test martingales consistently detect *only* sharp distribution drifts. Recall from Section 3.1 that increasing $\pi_1^T$ from 0.1 to 0.9 results in more than 20% accuracy drop. When a drift is slow (Figures 10a and 10b), conformal test martingales demonstrate much less power.

- Inspired by the ideas of Vovk et al. (2021) who assumed, like us, that (some) true data labels are observed, we designed a conformity score that reflects decrease in performance.

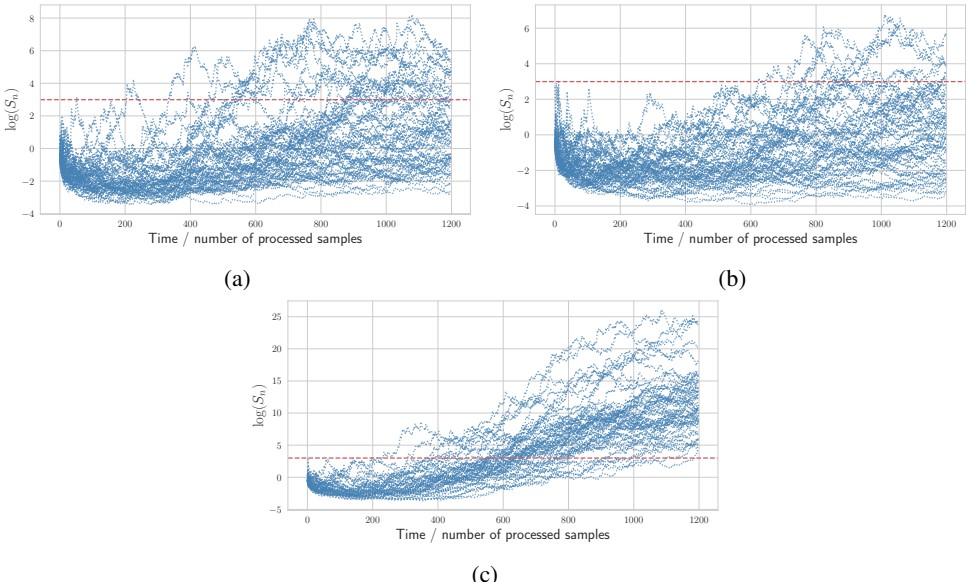

Figure 10: 50 runs of conformal test martingales (blue dotted lines) under gradual distribution drifts: (a) slow and benign, (b) slow and harmful, (c) sharp and harmful. The horizontal red dashed line outlines to the rejection threshold due to Ville's inequality. Note that conformal test martingales consistently detect only sharp distribution drifts. Moreover, conformal test martingales illustrate similar behavior under (a) and (b) but the corresponding settings are drastically different.

> On Figures 10a and 10b, conformal test martingales illustrate similar behavior but the corresponding settings are drastically different. Only one corresponds to a benign drift when the risk does not become significantly worse than the source risk. Thus, even though it is possible to make conformal test martingale reflect degrading performance, it is hard to incorporate evaluation of the malignancy of a change *in an interpretable way*, like a decrease of an important metric.

- Another problem of using conformal test martingales to be aware of is that after some time the corresponding values of the test martingale (larger implies more evidence for a shift) could start to decrease as the shifted distribution becomes the 'new normal' (Vovk et al., 2021). This is because they measure deviations from iid data, not degradations in performance from some benchmark (like source accuracy).

## B  LOSS FUNCTIONS

For our simulations, we consider the following bounded losses. Below, let $\widehat{y}(x; f) := \arg\max_{k \in \mathcal{Y}} f_k(x)$ denote the label prediction of a model $f$ on a given input $x \in \mathcal{X}$.

**Multiclass losses.** The most common example is arguably the misclassification loss and its generalization that allows for a label-dependent cost of a mistake:

$$\ell^{\text{mis}}(f(x), y) := \mathbb{1}\{\widehat{y}(x; f) \neq y\} \in \{0, 1\}, \quad \ell^{\text{w-mis}}(f(x), y) := \ell_y \cdot \mathbb{1}\{\widehat{y}(x; f) \neq y\} \in [0, L],$$

where $\{\ell_k\}_{k \in \mathcal{Y}}$ is a collection of per-class costs and $L = \max_{k \in \mathcal{Y}} \ell_k$. The loss $\ell^{\text{w-mis}}$ is more relevant to high-stakes decision settings and imbalanced classification. However, high accuracy alone can often be insufficient. The Brier score (squared error), introduced initially for the binary setting (Brier, 1950), is commonly employed to encourage calibration of probabilistic classifiers. For multiclass problems, one could consider the mean-squared error of the whole output vector:

$$\ell^{\text{brier}}(f(x), y) := \frac{1}{2}\|f(x) - h(y)\|^2 \in [0, 1], \tag{11}$$

where $h : \mathcal{Y} \to \{0, 1\}^{|\mathcal{Y}|}$ is a one-hot label encoder: $h_{y'}(y) = \mathbb{1}\{y' = y\}$ for $y, y' \in \mathcal{Y}$. Top-label calibration (Gupta & Ramdas, 2022) restricts attention to the entry corresponding to the top-ranked label. A closely related loss function, which we call the *top-label Brier score*, is the following:

$$\ell^{\text{brier-top}}(f(x), y) := (f_{\widehat{y}(x;f)}(x) - \mathbb{1}\{\widehat{y}(x;f) = y\})^2 = (f_{\widehat{y}(x;f)}(x) - h_{\widehat{y}(x;f)}(y))^2 \in [0, 1]. \quad (12)$$

Alternatively, instead of the top-ranked label, one could focus only on the entry corresponding to the true class. It gives rise to another loss function which we call the *true-class* Brier score:

$$\ell^{\text{brier-true}}(f(x), y) := (f_y(x) - 1)^2 \in [0, 1]. \quad (13)$$

The loss functions $\ell^{\text{brier}}$, $\ell^{\text{brier-top}}$, $\ell^{\text{brier-true}}$ trivially reduce to the same loss function in the binary setting. In Appendix C, we present a more detailed study of the Brier score in the multiclass setting with several illustrative examples.

**Set-valued predictors.** The proposed framework can be used for set-valued predictors that output a subset of $\mathcal{Y}$ as a prediction. Such predictors naturally arise in multilabel classification, where more than a single label can be the correct one, or as a result of post-processing point predictors. Post-processing could target covering the correct label of a test point with high probability (Vovk et al., 2005) or controlling other notions of risk (Bates et al., 2021) like the miscoverage loss:

$$\ell^{\text{miscov}}(y, S(x)) = \mathbb{1}\{y \notin S(x)\}, \quad (14)$$

where $S(x)$ denotes the output of a set-valued predictor on any given input $x \in \mathcal{X}$. When considering multilabel classification, relevant loss functions include the symmetric difference between the output and the set of true labels, false negative rate and false discovery rate.

## C  BRIER SCORE IN THE MULTICLASS SETTING

This section contains derivations of decompositions stated in Section 2 and comparisons between introduced versions of the Brier score.

**Brier score decompositions.** Define a function $c : \mathcal{X} \to \Delta^{|\mathcal{Y}|}$, with entries $c_k(X) := \mathbb{P}(Y = k \mid f(X))$. In words, coordinates of $c(X)$ represent the true conditional probabilities of belonging to the corresponding classes given an output vector $f(X)$. Recall that $h : \mathcal{Y} \to \{0, 1\}^{|\mathcal{Y}|}$ is a one-hot label encoder. The expected Brier score for the case when the whole output vector is considered (that is, the expected value of the loss defined in equation 11) satisfies the following decomposition:

$$
\begin{aligned}
2 \cdot R^{\text{brier}}(f) &= \mathbb{E}\|f(X) - h(Y)\|^2 \\
&= \mathbb{E}\|f(X) - c(X) + c(X) - h(Y)\|^2 \\
&\stackrel{(a)}{=} \mathbb{E}\|f(X) - c(X)\|^2 + \mathbb{E}\|c(X) - h(Y)\|^2 \\
&= \mathbb{E}\|f(X) - c(X)\|^2 + \mathbb{E}\|c(X) - \mathbb{E}[c(X)] + \mathbb{E}[c(X)] - h(Y)\|^2 \\
&\stackrel{(b)}{=} \underbrace{\mathbb{E}\|f(X) - c(X)\|^2}_{\text{calibration error}} - \underbrace{\mathbb{E}\|c(X) - \mathbb{E}[c(X)]\|^2}_{\text{sharpness}} + \underbrace{\mathbb{E}\|h(Y) - \mathbb{E}[h(Y)]\|^2}_{\text{intrinsic uncertainty}}.
\end{aligned}
$$

Above, (a) follows by conditioning on $f(X)$ for the cross-term and recalling that $\mathbb{E}[h(Y) \mid f(X)] = c(X)$, (b) also follows by conditioning on $f(X)$ and noticing that $\mathbb{E}[h(Y)] = \mathbb{E}[c(X)]$. Now, recall that a predictor is (canonically) calibrated if $f(X) \stackrel{a.s.}{=} c(X)$, in which case the calibration error term is simply zero.

Next, we consider the top-label Brier score $\ell^{\text{brier-top}}$. Define $c^{\text{top}} : \mathcal{X} \to [0, 1]$, as:

$$c^{\text{top}}(X) := \mathbb{P}\left(Y = \widehat{y}(X; f) \mid f_{\widehat{y}(X;f)}(X), \widehat{y}(X; f)\right),$$

or the fraction of correctly classified points among those that are predicted to belong to the same class and share the same confidence score as $X$. Following essentially the same argument as for the

standard Brier score, we get that:

$$
\begin{aligned}
&\mathbb{E}\left[\ell^{\text{brier-top}}(f(X), Y)\right] \\
&= \mathbb{E}\left(f_{\widehat{y}(X;f)}(X) - h_{\widehat{y}(X;f)}(Y)\right)^2 \\
&= \mathbb{E}\left(f_{\widehat{y}(X;f)}(X) - c^{\text{top}}(X) + c^{\text{top}}(X) - h_{\widehat{y}(X;f)}(Y)\right)^2 \\
&= \mathbb{E}\left(f_{\widehat{y}(X;f)}(X) - c^{\text{top}}(X)\right)^2 + \mathbb{E}\left(c^{\text{top}}(X) - h_{\widehat{y}(X;f)}(Y)\right)^2 \\
&= \underbrace{\mathbb{E}\left(f_{\widehat{y}(X;f)}(X) - c^{\text{top}}(X)\right)^2}_{\text{top-label calibration error}} - \underbrace{\mathbb{E}\left(c^{\text{top}}(X) - \mathbb{E}\left[c^{\text{top}}(X)\right]\right)^2}_{\text{top-label sharpness}} + \underbrace{\text{Var}\left(h_{\widehat{y}(X;f)}(Y)\right)}_{\substack{\text{variance of the} \\ \text{misclassification loss}}}.
\end{aligned}
$$

Note that in contrast to the classic Brier score decomposition, the last term in this decomposition depends only on the top-class prediction of the underlying predictor, and thus on its accuracy.

**Comparison of the scores in multiclass setting.** Recall that the difference between three versions of the Brier score arises when one moves beyond the binary classification setting. We illustrate the difference by considering a 4-class classification problem where the data represent a balanced (that is all classes are equally likely) mixture of 4 Gaussians with identity covariance matrix and mean vectors being the vertices of a 2-dimensional unit cube. One such sample is presented on Figure 11a.

Next, we analyze locally the Brier scores when the Bayes-optimal rule is used as an underlying predictor, that is we split the area into small rectangles and estimate the mean score within each rectangle by a sample average. The results are presented on Figures 11b, 11c and 11d. Note that the difference between the assigned scores is mostly observed for the points that lie at the intersection of 4 classes where the support of the corresponding output vectors is large.

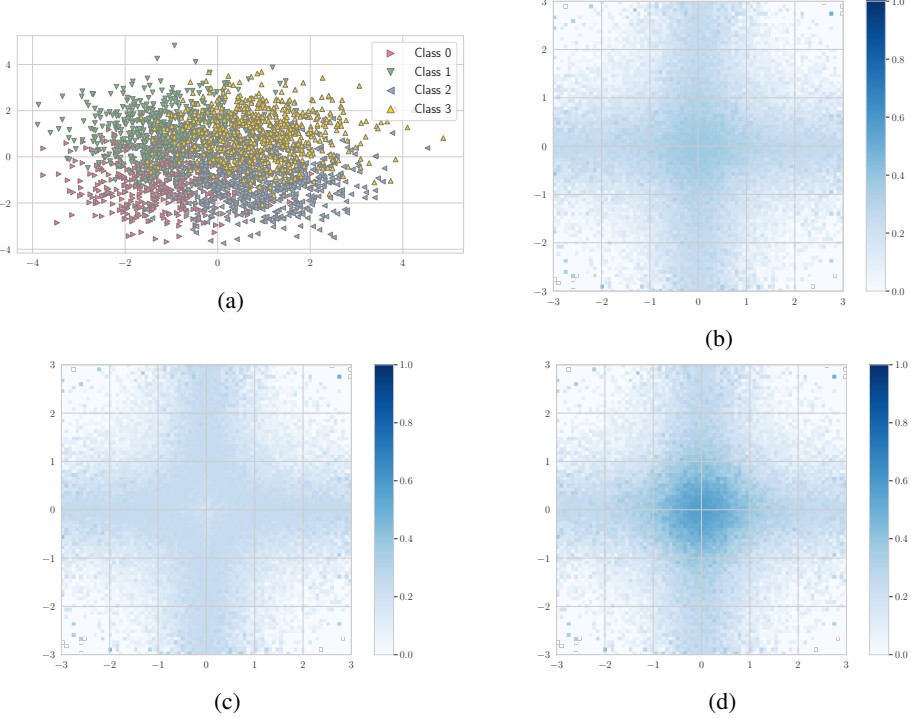

Figure 11: (a) Visualization of 4-class classification problem with all classes being equally likely; (b) localized classic Brier score $\ell^{\text{brier}}$ (equation 11); (c) localized top-label Brier score $\ell^{\text{brier-top}}$ (equation 12); (d) localized true-class Brier score $\ell^{\text{brier-true}}$ (equation 13).

**Brier scores under label shift.** Here we consider the case when label shift on the target domain is present. First, introduce the label likelihood ratios, also known as the importance weights, $w_y :=$

$\pi_y^T/\pi_y^S$, $y \in \mathcal{Y}$. For measuring the strength of the shift, we introduce the *condition number*: $\kappa = \sup_y w_y / \inf_{y:w_y \neq 0} w_y$. Note that when the shift is not present, the condition number $\kappa = 1$. To evaluate the sensitivity of the losses to the presence of label shift, we proceed as follows: first, the class proportions for both source and target domains are sampled from the Dirichlet distribution (to avoid extreme class proportion, we perform truncation at levels 0.15 and 0.85 and subsequent renormalization). Then we use the Bayes-optimal rule for the source domain to perform predictions on the target and compute the corresponding losses. On Figure 12, we illustrate relative increase in the average Brier scores plotted against the corresponding condition number when all data points are considered (Figure 12a) and when attention is restricted to the area where classes highly intersect (Figure 12b). In general, all three versions of the Brier score suffer similarly on average, but in the localized area where classes highly intersect, the top-label Brier score does not increase significantly under label shift.

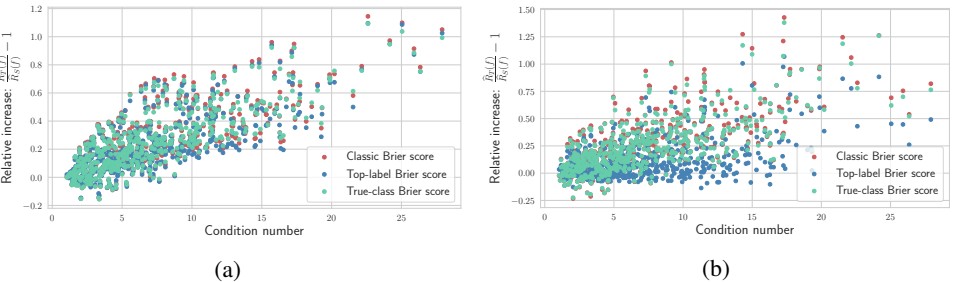

(a)                                         (b)

Figure 12: (a) Relative increase for different versions of the Brier score in the multiclass setting under label shift; (b) Relative increase for different versions of the Brier score in the multiclass setting under label shift when attention is restricted to the area where classes highly intersect (cube with vertices at $(\pm1/2, \pm1/2)$). While in general, all three versions of the Brier score suffer similarly on average, in the localized area where classes highly intersect, the top-label Brier score does not increase significantly under label shift.

## D PROOFS

*Proof of Proposition 1.* For brevity, we omit writing $f$ for the source/target risks and the corresponding bound upper/lower confidence bounds. Starting with the absolute change and the null $H_0 : R_T - R_S \leq \varepsilon_{\text{tol}}$, we have:

$$\mathbb{P}_{H_0}\left(\exists t \geq 1 : \widehat{L}_T^{(t)} > \widehat{U}_S + \varepsilon_{\text{tol}}\right)$$
$$= \mathbb{P}_{H_0}\left(\exists t \geq 1 : \left(\widehat{L}_T^{(t)} - R_T\right) - \left(\widehat{U}_S - R_S\right) > \varepsilon_{\text{tol}} - (R_T - R_S)\right)$$
$$\leq \mathbb{P}_{H_0}\left(\exists t \geq 1 : \left(\widehat{L}_T^{(t)} - R_T\right) - \left(\widehat{U}_S - R_S\right) > 0\right).$$

Note that $\exists t \geq 1 : (\widehat{L}_T^{(t)} - R_T) - (\widehat{U}_S - R_S) > 0$ implies that either $\exists t \geq 1 : \widehat{L}_T^{(t)} - R_T > 0$ or $\widehat{U}_S - R_S < 0$. Thus, invoking union bound yields:

$$\mathbb{P}_{H_0}\left(\exists t \geq 1 : \left(\widehat{L}_T^{(t)} - R_T\right) - \left(\widehat{U}_S - R_S\right) > 0\right)$$
$$\leq \mathbb{P}\left(\exists t \geq 1 : \widehat{L}_T^{(t)} - R_T > 0\right) + \mathbb{P}\left(\widehat{U}_S - R_S < 0\right)$$
$$\leq \delta_T + \delta_S,$$

by construction and validity guarantees for $\widehat{L}_T^{(t)}$ and $\widehat{U}_S$. Similarly, considering the relative change, i.e., the null: $H_0' : R_T \leq (1 + \varepsilon'_{\text{tol}})R_S$, we have:

$$\mathbb{P}_{H_0'}\left(\exists t \geq 1 : \widehat{L}_T^{(t)} > (1 + \varepsilon'_{\text{tol}})\widehat{U}_S\right)$$
$$= \mathbb{P}_{H_0'}\left(\exists t \geq 1 : \left(\widehat{L}_T^{(t)} - R_T\right) - (1 + \varepsilon'_{\text{tol}})\left(\widehat{U}_S - R_S\right) > (1 + \varepsilon'_{\text{tol}})R_S - R_T\right)$$
$$\leq \mathbb{P}_{H_0'}\left(\exists t \geq 1 : \left(\widehat{L}_T^{(t)} - R_T\right) - (1 + \varepsilon'_{\text{tol}})\left(\widehat{U}_S - R_S\right) > 0\right).$$

Similarly, note that $\exists t \geq 1 : (\widehat{L}_T^{(t)} - R_T) - (1 + \varepsilon'_{\text{tol}})(\widehat{U}_S - R_S) > 0$ implies that either $\exists t \geq 1 :$ $\widehat{L}_T^{(t)} - R_T > 0$ or $\widehat{U}_S - R_S < 0$. Thus, invoking union bound yields the desired result.

$\square$

## E   PRIMER ON THE UPPER AND LOWER CONFIDENCE BOUNDS

This section contains the details for the concentration results used in this work. Results presented in this section are not new were developed in a series of recent works (Waudby-Smith & Ramdas, 2021; Howard et al., 2021). We follow the notation from Waudby-Smith & Ramdas (2021) for consistency and use the superscript $(t)$ when referring to confidence sequences (CS) and $(n)$ when referring to confidence intervals (CI).

**Predictably-mixed Hoeffding's (PM-H) confidence sequence.**   Then the upper and lower endpoints of the predictably-mixed Hoeffding's (PM-H) confidence sequence are given by:

$$L_{\text{PM-H}}^{(t)} := \left( \frac{\sum_{i=1}^t \lambda_i Z_i}{\sum_{i=1}^t \lambda_i} - \frac{\log(1/\delta) + \sum_{i=1}^t \psi_H(\lambda_i)}{\sum_{i=1}^t \lambda_i} \right),$$

$$U_{\text{PM-H}}^{(t)} := \left( \frac{\sum_{i=1}^t \lambda_i Z_i}{\sum_{i=1}^t \lambda_i} + \frac{\log(1/\delta) + \sum_{i=1}^t \psi_H(\lambda_i)}{\sum_{i=1}^t \lambda_i} \right),$$

where $\psi_H(\lambda) := \lambda^2/8$ and $\lambda_1, \lambda_2, \ldots$ is a predictable mixture. We use a particular predictable mixture given by:

$$\lambda_t^{\text{PM-H}} := \sqrt{\frac{8 \log(1/\delta)}{t \log(t+1)}} \wedge 1.$$

When approximating the risk on the source domain, one would typically have a holdout sample of a fixed size $n$, and so one could either use the classic upper limit of the Hoeffding's confidence interval, which is recovered by taking equal $\lambda_i = \lambda = \sqrt{8 \log(1/\delta)/n}$, $i = 1, \ldots, n$, in which case the upper and lower limits simplify to:

$$L_{\text{H}}^{(n)} := \left( \frac{\sum_{i=1}^n Z_i}{n} - \sqrt{\frac{\log(1/\delta)}{2n}} \right), \quad U_{\text{H}}^{(n)} := \left( \frac{\sum_{i=1}^n Z_i}{n} + \sqrt{\frac{\log(1/\delta)}{2n}} \right),$$

or by considering running intersection of the predictably mixed Hoeffding's confidence sequence: $(\min_{t \leq n} U_{\text{PM-H}}^{(t)}, \max_{t \leq n} L_{\text{PM-H}}^{(t)})$.

**Predictably-mixed empirical-Bernstein (PM-EB) confidence sequence.**   The upper and lower endpoints of the predictably-mixed empirical-Bernstein (PM-EB) confidence sequence are given by:

$$L_{\text{PM-EB}}^{(t)} := \frac{\sum_{i=1}^t \lambda_i Z_i}{\sum_{i=1}^t \lambda_i} - \frac{\log(1/\delta) + \sum_{i=1}^t v_i \psi_E(\lambda_i)}{\sum_{i=1}^t \lambda_i},$$

$$U_{\text{PM-EB}}^{(t)} := \frac{\sum_{i=1}^t \lambda_i Z_i}{\sum_{i=1}^t \lambda_i} + \frac{\log(1/\delta) + \sum_{i=1}^t v_i \psi_E(\lambda_i)}{\sum_{i=1}^t \lambda_i},$$

where

$$v_i := 4 \left( X_i - \widehat{\mu}_{i-1} \right)^2, \quad \text{and} \quad \psi_E(\lambda) := \left( -\log(1 - \lambda) - \lambda \right)/4, \quad \text{for} \quad \lambda \in [0, 1).$$

One particular choice of a predictable mixture $(\lambda_t^{\text{PM-EB}})_{t=1}^\infty$ is given by:

$$\lambda_t^{\text{PM-EB}} := \sqrt{\frac{2 \log(1/\delta)}{\widehat{\sigma}_{t-1}^2 t \log(1+t)}} \wedge c, \quad \widehat{\sigma}_t^2 := \frac{\frac{1}{4} + \sum_{i=1}^t (Z_i - \widehat{\mu}_i)^2}{t+1}, \quad \widehat{\mu}_t := \frac{\frac{1}{2} + \sum_{i=1}^t Z_i}{t+1},$$

for some $c \in (0, 1)$. We use $c = 1/2$ and also set $\widehat{\mu}_0 = 1/2$, $\widehat{\sigma}_0 = 1/4$. If given a sample of a fixed size $n$, we consider running intersection along with the predictable sequence given by:

$$\lambda_t^{\text{PM-EB}} = \sqrt{\frac{2 \log(1/\delta)}{n \widehat{\sigma}_{t-1}^2}} \wedge c, \quad t = 1, \ldots, n.$$

**Betting-based confidence sequence.** Tighter confidence intervals/sequences can be obtained by invoking tools from martingale analysis and deploying betting strategies for confidence intervals/sequences construction proposed in (Waudby-Smith & Ramdas, 2021). While those can not be computed in closed-form, empirically they tend to outperform previously considered confidence intervals/sequences. Recall that we are primarily interested in one-sided results, and for simplicity we discuss. For any $m \in [0, 1]$, introduce a capital (wealth) process:

$$\mathcal{K}_t^{\pm}(m) := \prod_{i=1}^{t} \left( 1 \pm \lambda_i^{\pm}(m) \cdot (Z_i - m) \right),$$

where $\left\{\lambda_t^+(m)\right\}_{t=1}^{\infty}$ and $\left\{\lambda_t^-(m)\right\}_{t=1}^{\infty}$ are $[0, 1/m]$-valued and $[0, 1/(1-m)]$-valued predictable sequences respectively. A particular example of such predictable sequences we use is given by:

$$\lambda_t^+(m) := \left|\dot{\lambda}_t^+\right| \wedge \frac{c}{m}, \quad \lambda_t^-(m) := \left|\dot{\lambda}_t^-\right| \wedge \frac{c}{1-m},$$

where, for example, $c = 1/2$ or $3/4$ and $\dot{\lambda}_t^{\pm}$ do not depend on $m$. Such choice guarantees, in particular, that the resulting martingale is nonnegative. For example, the wealth process $\mathcal{K}_t^+(m)$ incorporates a belief that the true mean $\mu$ is larger than $m$ and the converse belief is incorporated in $\mathcal{K}_t^-(m)$, that is, the wealth is expected to be accumulated under the corresponding belief (e.g., consider $m = 0$ and the corresponding $\mathcal{K}_t^+(0)$ with a high value, and $m = 1$ and the corresponding $\mathcal{K}_t^+(1)$ with a low value). Using that $\mathcal{K}_t^+(m)$ is non-increasing in $m$, i.e., $m_2 \geq m_1$, then $\mathcal{K}_t^+(m_2) \leq \mathcal{K}_t^+(m_1)$, we thus can use grid search (up to specified approximation error $\Delta_{\text{grid}}$) to efficiently approximate $L_{\text{Bet}}^{(t)} = \inf B_t^+$, where

$$\mathcal{B}_t^+ := \left\{ m \in [0, 1] : \mathcal{K}_t^+(m) < 1/\delta \right\},$$

that is, the collection of all $m$ for which that the capital wasn't accumulated. Then we can consider $L_{\text{Bet}}^{(n)} = \max_{t \leq n} L_{\text{Bet}}^{(t)}$. When $m = \mu$ is considered, none of $\mathcal{K}_t^{\pm}(\mu)$ is expected to be large, since by Ville's inequality:

$$\mathbb{P}\left(\exists t \geq 1 : \mathcal{K}_t^+(\mu) \geq 1/\delta\right) \leq \delta,$$

and thus we know that with high probability the true mean is larger than $L_{\text{Bet}}^{(n)}$. That is,

$$\mathbb{P}\left(\mu < L_{\text{Bet}}^{(n)}\right) = \mathbb{P}\left(\mu < \max_{t \leq n} L_{\text{Bet}}^{(t)}\right) = \mathbb{P}\left(\exists t \geq 1 : \mu < \inf B_t^+\right)$$
$$= \mathbb{P}\left(\exists t \geq 1 : \mathcal{K}_t^+(\mu) \geq 1/\delta\right) \leq \delta.$$

By a similar argument, we get that with high probability, the true mean is less than $U_{\text{Bet}}^{(n)} = \min_{t \leq n} \sup B_t^-$:

$$\mathbb{P}\left(\mu > U_{\text{Bet}}^{(n)}\right) = \mathbb{P}\left(\mu > \min_{t \leq n} \sup B_t^-\right) = \mathbb{P}\left(\exists t \geq 1 : \mu > \sup B_t^-\right)$$
$$= \mathbb{P}\left(\exists t \geq 1 : K_t^-(\mu) \geq 1/\delta\right) \leq \delta.$$

**Conjugate-mixture empirical-Bernstein (CM-EB) confidence sequence.** Below, we present a shortened description of CM-EB and refer the reader to Howard et al. (2021) for more details. Assume that one observes a sequence of random variables $Z_t$, bounded in $[a, b]$ almost surely for all $t$, and the goal is to construct a confidence sequence for $\mu_t := t^{-1} \sum_{i=1}^{t} \mathbb{E}_{i-1} Z_i$, the average conditional expectation. Theorem 4 in Howard et al. (2021) states that for any $(\widehat{Z}_t)$, $[a, b]$-valued predictable sequence, and any $u$, the sub-exponential uniform boundary with crossing probability $\alpha$ for scale $c = b - a$, it holds that:

$$\mathbb{P}\left(\forall t \geq 1 : \left|\overline{Z}_t - \mu_t\right| < \frac{u\left(\sum_{i=1}^{t}(Z_i - \widehat{Z}_i)^2\right)}{t}\right) \geq 1 - 2\alpha,$$

where a reasonable choice for the predictable sequence $(\widehat{Z}_t)$ is given by $\widehat{Z}_t = (t-1)^{-1} \sum_{i=1}^{t-1} Z_i$.

The key challenge which is addressed by conjugate mixtures is obtaining sublinear uniform boundaries that allows the radius of the confidence sequences to shrink to zero asymptotically. When a closed form of the confidence sequence is not required, the gamma-exponential mixture generally yields the tightest bounds. The procedure relies on the following *mixing* result (Lemma 2, Howard et al. (2021)) which states that for any $\alpha \in (0,1)$ and any chosen probability distribution $F$ on $[0, \lambda_{\max})$:

$$u_\alpha^{\mathrm{CM}}(v) := \sup \left\{ s : \in \mathbb{R} : \underbrace{\int \exp\left(\lambda s - \psi(\lambda) v\right) \mathrm{d}F(\lambda)}_{=:m(s,v)} < \frac{1}{\alpha} \right\},$$

yields a sub-$\psi$ uniform boundary with crossing probability $\alpha$. When the gamma distribution is used for mixing, $m(s,v)$ has a closed form given in [Proposition 9, Howard et al. (2021)]. Subsequently, the resulting *gamma-exponential mixture* boundary $u_\alpha^{\mathrm{CM}}(v)$ is computed by numerically solving the equation $m(s,v) = 1/\alpha$ in $s$. Howard et al. (2021) provide the packages for computing the corresponding confidence sequence.

## F  EXPERIMENTS ON SIMULATED DATA

Figure 13 illustrates data samples for two settings where the presence of label shift is not expected to cause degradation in model performance (measured in terms of absolute increase in misclassification risk) for the first one ($\mu_0 = (-2,0)^\top$, $\mu_1 = (2,0)^\top$), but label shift may potentially degrade performance for the second ($\mu_0 = (-1,0)^\top$, $\mu_1 = (1,0)^\top$). For both cases, samples follow the same data generation pipeline as in Section 1 with only changes in class centers.

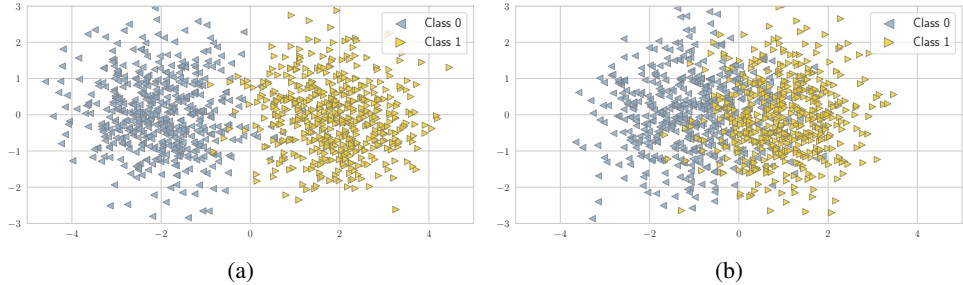

|     (a)     |     (b)     |

Figure 13: (a) Simulated dataset with well-separated classes. Presence of label shift presumably will not lead to a high absolute increase in the misclassification risk. (b) In contrast, when the classes are *not* well-separated, presence of label shift presumably might hurt the misclassification risk.

### F.1  BRIER SCORE AS A TARGET METRIC

Here we replicate the empirical study from Section 3.1 but use the Brier score as a target metric. Recall that all three multiclass versions of the Brier score discussed in this work reduce to the same loss in the binary setting. First, we compare upper confidence bounds for the Brier score computed by invoking different concentration results on Figure 14. Similar to the misclassification risk, variance-adaptive confidence bounds exploit the low-variance structure and are tighter when compared against the non-adaptive one.

Next, we perform empirical analysis of the power of the testing framework. We take $\varepsilon_{\mathrm{tol}} = 0.1$ which corresponds to testing for a 10% relative increase in the Brier score. We take $n_S = 1000$ data points from the source distribution to compute upper confidence bound on the source risk $\widehat{U}_S(f)$. Subsequently, we sample the data from the target distribution in batches of 50, with maximum number of samples from the target set to be 2000. On Figure 14b, we present the proportion of cases when the null hypothesis is rejected out of 250 simulations performed for each candidate class 1 probability. On Figure 14c, we illustrate average sample size from the target domain that was needed to reject the null hypothesis. When a stronger and more harmful label shift is present, less samples are required to reject the null, and moreover, the most powerful tests utilize upper/lower

confidence bounds obtained via the betting approach. On Figure 14d, we present the comparison of different time-uniform lower confidence bounds.

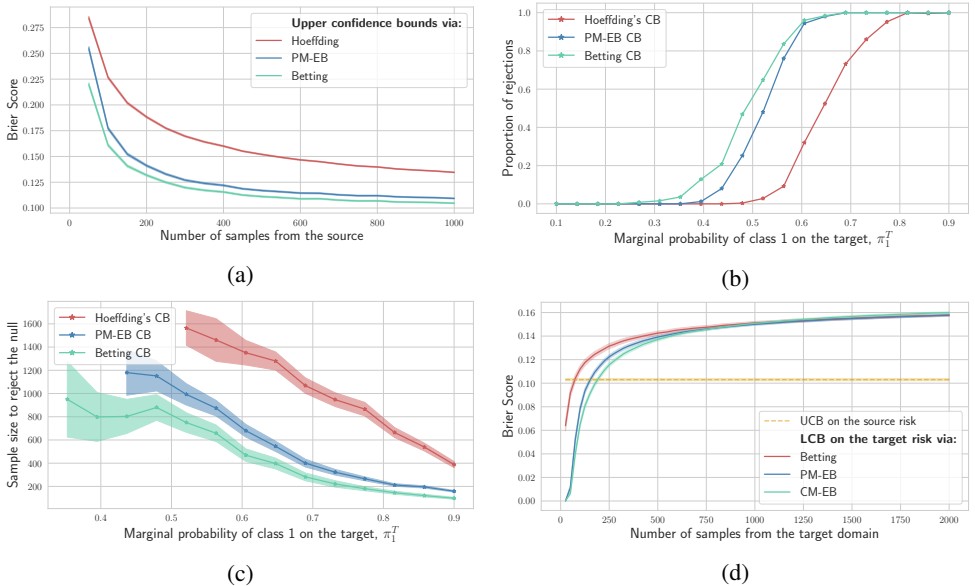

Figure 14: (a) Upper confidence bounds $\widehat{U}_S(f)$ on the Brier score for the source domain. Similar to the misclassification risk, variance-adaptive confidence bounds are tighter when compared against the Hoeffding's one. For each fixed number of data points from the source domain used to compute $\widehat{U}_S(f)$, presented results are aggregated over 1000 random data draws. (b) Proportion of null rejections made by the procedure when testing for 10% relative increase of the Brier score. (c) Average sample size from the target distribution that was needed to reject the null. Invoking tighter concentration results allows to raise an alarm after processing less samples from the target domain. (d) Different lower/upper confidence bounds on the target/source domain for the Brier score.

# G EXPERIMENTS ON REAL DATASETS

## G.1 MNIST-C SIMULATION

**Architecture and training.** For MNIST-C dataset, we train a shallow CNN with two convolutional layers (each with $3 \times 3$ kernel matrices), each followed by max-pooling layers. Subsequently, the result is flattened and followed by a dropout layer ($p = 0.5$), a fully-connected layer with 128 neurons and an output layer. Note that the network is trained on original (clean) MNIST data, which is split split into two folds with 10% of data used for validation purposes. All images are scaled to $[0, 1]$ range before the training is performed.

**Training multiple networks.** To validate observations regarding shift malignancy from Section 3.2, we train 5 different networks (following the same training protocol) and report aggregated (over 25 random ordering of the data from the target) results on Figure 15. The observation that applying translation to the MNIST images represents a harmful shift is consistent across all networks.

## G.2 CIFAR-10-C SIMULATION

**Architecture and training.** The model underlying a set-valued predictor is a standard ResNet-32. It is trained for 50 epochs on the original (clean) CIFAR-10 dataset, without data augmentation, using 10% of data for validation purposes. All images are scaled to $[0, 1]$ range before the training is performed. The accuracy of the resulting network is $\approx 80.5\%$.

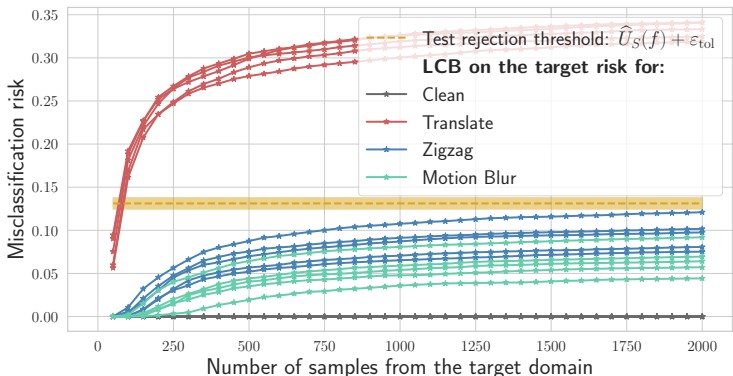

Figure 15: Lines of the same color correspond to 5 different CNNs. For each network, the results aggregated over 25 random runs of the testing framework for randomly permuted test data. Applying translate effect is consistently harmful to the performance of CNNs trained on clean MNIST data. The bar around the yellow dashed line corresponds to 2 standard deviations.

**Transforming a point predictor into a set-valued one.** To transform a point predictor into a set-valued one, we consider a sequence of candidate prediction sets $S_\lambda(x)$, parameterized by univariate parameter $\lambda$, with larger $\lambda$ leading to larger prediction sets. Under the considered setting, the underlying predictor is an estimator of the true conditional probabilities $\pi_y(x) = \mathbb{P}(Y = y \mid X = x)$. Given a learnt predictor $f$, we can define

$$\rho_y(x; f) := \sum_{k=1}^{K} f_k(x) \cdot \mathbb{1}\{f_k(x) > f_y(x)\}$$

to represent estimated probability mass of the labels that are more likely than $y$. Subsequently, we can consider the following sequence of set-valued predictors:

$$S_\lambda(x) = \{y \in \mathcal{Y} : \rho_y(x; f) \leq \lambda\}, \quad \lambda \in \Lambda := [0, 1],$$

that is the sequence is based on the estimated density level sets, starting by including the most likely labels according to the predictions of $f$. To tune the parameter $\lambda$, we follow Bates et al. (2021): we keep a labeled holdout *calibration set*, and use it to pick:

$$\widehat{\lambda} = \inf\left\{\lambda \in \Lambda : \ \widehat{R}^+(\lambda') < \beta, \ \forall \lambda' > \lambda\right\},$$

where $\widehat{R}^+(\lambda')$ is an upper confidence bound for the risk function at level $\beta$. The resulting set-valued predictor is then $(\beta, \gamma)$-RCPS, that is,

$$\mathbb{P}\left(R(S_{\widehat{\lambda}}) \leq \beta\right) \geq 1 - \gamma,$$

under the i.i.d. assumption. More details and validity guarantees can be found in Bates et al. (2021).

**Set-valued predictor when $\beta = 0.05$ is used as a prescribed error level.** In contrast to $\beta = 0.1$ used in the main paper, we also consider decreasing $\beta$ to $0.05$, which in words, corresponds to increasing a desired coverage level of the resulting set-valued predictor. Figure 16a compares average sizes of the prediction sets for two candidate values: $\beta_1 = 0.05$ and $\beta_2 = 0.1$, when the set-valued predictor is passes either clean CIFAR-10 data, or images to which fog corruption has been applied. As expected, decreasing $\beta$ leads to larger prediction sets on average, with the size increasing when corrupted images are passes as input, that the size reflects uncertainty in prediction. In Figure 16b, we observe that when we run the testing framework for the set-valued predictor corresponding to $\beta = 0.05$, only the most severe version of corruptions by adding fog is consistently marked as harmful, and thus raising an alarm. Similar to Section 3.2, we also use $\varepsilon_{\text{tol}} = 0.05$.

## H  TESTING FOR HARMFUL COVARIATE SHIFT

In this section, we consider a case when covariate shift is present on the target domain, that is, when the marginal distribution $P(X)$ changes but $P(Y|X)$ does not. Consider the binary classification

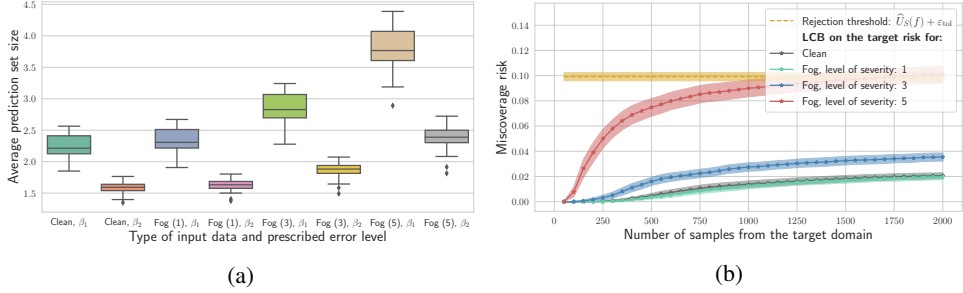

(a)                                                                      (b)

Figure 16: (a) Average size of prediction sets for $\beta_1 = 0.05$ and $\beta_2 = 0.1$ and different types of input data. First, lower $\beta$, corresponding to higher desired coverage, leads to larger prediction sets on average. Second, average size of the prediction sets increases when more corrupted images are passed as input, thus reflecting uncertainty in prediction. (b) Results of running the framework when $\beta_1 = 0.05$ is used to construct a wrapper. Observe that setting a lower prescribed error level $\beta$ and thus enlarging resulting prediction sets partially mitigates the impact of corrupting images with the fog effect. However, the most severe form of such corruption still consistently leads to rejecting the null. The bars around dashed and solid lines correspond to 2 standard deviations.

setting with accuracy being a target metric. It is known that the optimal decision rule for this case is the Bayes-optimal rule:

$$f^\star(x) = \mathbb{1}\left\{\mathbb{P}\left(Y = 1 \mid X = x\right) \geq 1/2\right\},$$

which minimizes the probability of misclassifying a new data point. Then one might expect that a change in the marginal distribution of $X$ does not necessarily call for retraining if a learnt predictor $f$ is 'close' to $f^\star$ (which itself could be a serious assumption to make). However, a change in the marginal distribution of $X$ could indicate, in particular, that one should reconsider certain design choices made during training a model. To illustrate when it could be useful, we consider the following data generation pipeline:

1. Initially, each point is assigned either to the origin or to a circle of radius 1 centered at the origin with probability $1/2$.

2. For points assigned to the origin, the coordinates are sampled from multivariate Gaussian distribution with zero mean and rescaled identity covariance matrix: $\frac{1}{36}I_2$.

3. For points assigned to the circle, the coordinates are sampled from multivariate Gaussian distribution with the same covariance matrix but with the mean vector:
$$\begin{cases} \mu_x = \cos(\varphi), \\ \mu_y = \sin(\varphi), \end{cases}$$

   where $\varphi^S \sim \mathrm{Unif}([-\pi/3, \pi/3])$ on the source domain and $\varphi^T \sim \mathrm{Unif}([0, 2\pi])$ on the target domain (see Figure 17 for a visualization).

4. Then points are assigned the corresponding labels according to:
$$\mathbb{P}\left(Y = 1 \mid X = x\right) = \mathbb{1}\left\{x_1^2 + x_2^2 \geq \frac{1}{2}\right\}.$$

It is easy to see that a linear predictor, e.g., logistic regression, can achieve high accuracy if deployed on the data sampled from the source distribution. However, it will clearly fail to recover the true relationship between the features and responses. In this case, a change in the marginal distribution of $X$ might indicate that updating a functional class could be necessary. On this data, we also run the proposed framework testing for a 10% increase in misclassification risk ($\varepsilon_{\mathrm{tol}} = 0.1$). At each run, we use 200 points to train a logistic regression and 100 points to estimate the betting-based upper confidence bound on the source risk. On the target domain, we use the lower confidence bound due to conjugate-mixture empirical-Bernstein (CM-EB). The results presented on Figure 17c (which have been aggregated over 100 random data draws) illustrate that the framework successfully detects a harmful shift, requiring only a small number of samples to do so.

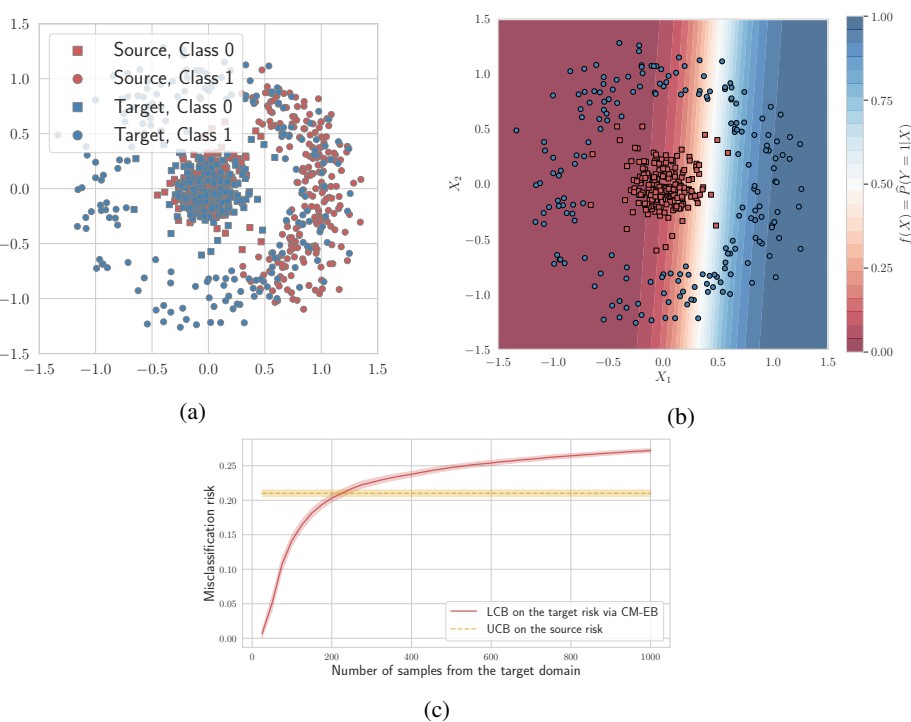

(a)

(b)

(c)

Figure 17: (a) Data samples from the source (red) and target (blue) distributions for the covariate shift simulation. (b) Logistic regression predictor learnt on the source distribution plotted along with a data sample from the target distribution. While learnt predictor clearly has high accuracy on the source domain, it fails to approximate the true underlying data generating distribution. (c) Results of running the framework when testing for a 10% increase in the misclassification risk. The framework detects a harmful shift after processing only a small number of samples.

