# OpenReview forum: "Tracking the risk of a deployed model and detecting harmful distribution shifts"
_ICLR.cc/2022/Conference — ICLR 2022 Poster_

### Official Review · Reviewer_a6Wo · 2021-10-26

**Correctness:** 4
**Technical Novelty And Significance:** 3
**Empirical Novelty And Significance:** 3
**Recommendation:** 6
**Confidence:** 2

**Main Review:**

**Strengths**

1. This paper has studied distribution drift in both i.i.d. and independence settings, where the non- i.i.d. assumption is more realistic and provide stronger results: since in real situation the distribution is more likely to drift gradually instead of a sharp change.

2. The sequential detection method is an intuitive way to continuously monitor model performance. In addition, the authors have also discussed several variants of multiclass losses to be combined with the sequential detection framework, which is commonly used in a variety of tasks.

**Weakness**

1. [Experimental results] All reported results of methods and baselines are proposed in the paper, e.g., Hoeffding, PM-EB, and betting are methods with different confidence bounds. There should be a direct comparison with other baselines and non-sequential methods, or at least show the behavior of a state-of-the-art approach to demonstrate what's the advantage of sequential testing. In addition, it is better to conduct both label and covariate shift experiments in sections 3.1 and 3.2.

2. [Notations] The authors first defined $Z$ in section 2.2 but it first appears in section 2.1; the definition of $R^{(t)} (f)$, what is the expectation over?

3. [Others] In section 1, the authors have mentioned current methods require assumptions on both covariate and label shift. Although the proposed method doesn't need this assumption, it is great to showcase how to overcome this issue when the assumption is violated.


**Summary Of The Paper:**

This paper proposed a sequential testing method for tracking distribution shifts for deployed models. The main idea is to utilize sequential hypothesis testing on a given objective function in a range of time stamps. The authors have studied a variety of different confidence intervals/sequences that certifies benign or harmful distribution drifts and have also demonstrated theoretical support for their claims. Empirical results on both simulated and real data have shown this sequential testing method is effective in detecting severe distribution drifts.

**Summary Of The Review:**

This paper proposes an effective and theoretically sound method for sequentially detecting harmful distribution drifts. Overall, this work is solid but still requires some improvements on the experiment.

## Update after Rebuttal

We thank the authors for their detailed responses and for updating the manuscripts. The authors have addressed my questions, but we tend to keep the current score based on the significance of the improvements made.

---

> ### Author Response · Authors · 2021-11-21
> **Response to Reviewer a6Wo**
>
> We warmly thank the reviewer for constructive and actionable feedback on the paper. Below, we address certain points that have been raised in the review. We hope that the reviewer may consider increasing their score if many/most of the concerns were reasonably addressed.
>
> You mentioned
>
> > **(joint answer)** (a) All reported results of methods and baselines are proposed in the paper. There should be a direct comparison with other baselines and non-sequential methods, or at least show ... what's the advantage of sequential testing. (b) In section 1, the authors have mentioned current methods require assumptions on both covariate and label shift. Although the proposed method doesn't need this assumption, it is great to showcase how to overcome this issue when the assumption is violated.
>
> To recap, to the best of our knowledge, all related works on testing for distribution shifts are either (i) non-sequential, or (ii) non-adaptive to the nature/malignancy of the shift.
>
> Regarding (i), adapting traditional (fixed-time) testing to the sequential setting would require performing corrections for multiple testing (ie multiple views of the data, at different timepoints); if the correction is not performed, validity is lost (false positives will abound) but if such a correction is performed, the test will be too conservative, due to the dependence among the tests being ignored (and not being taken advantage of). However, performing testing via sequential estimation allows to accumulate evidence over time, without throwing away any data or the necessity of performing corrections for multiple testing (these are implicitly handled efficiently using martingale methods that are underlying the sequential estimation procedures). To illustrate the shortcomings of deploying fixed-time tests under continuous monitoring, we performed a series of new experiments that are presented in Appendix A.1 of the updated version of the paper.
>
> Regarding (ii), we only know of one approach for performing sequential testing for the presence of a distribution shift/drift (Vovk et al., 2021). However, their approach requires certain design choices (choice of a conformal score function) to be made before it becomes fully practically relevant, but more importantly, it fails to account for the malignancy of a shift. We have included a detailed study of conformal test martingales in Appendix A.2 of the updated version of the paper. Like us, they also require access to true labels. The main shortcoming of deploying conformal test martingales (CTMs) include:
>
> - **Possibly low power:** only extreme (severe and sharp) shifts/drifts are consistently detected.
>
> - **Poor differentiation between harmful and benign cases:** it is hard to incorporate evaluation of the malignancy of a change in an interpretable way.
>
> - **Saturation:** values of CTMs (larger value implies more evidence for a shift) could saturate and then start to decrease after some time. This is because they measure deviations from iid data, not degradations in performance from some benchmark, like source accuracy; after some time, the shifted distribution becomes the ''new normal'', and is no longer treated as a shift by the method, even if the accuracy is now worse than it was at the start.
>
> You also mentioned
>
> > In addition, it is better to conduct both label and covariate shift experiments in sections 3.1 and 3.2.
>
> Since the label shift experiments were already in the paper, we presume that the reviewer is asking for only an additional covariate shift experiment. In the updated version of the paper, we now also include an experiment where on the target domain covariate shift is present. Due to space limits, it is presented in Appendix H with a proper referencing from the main part of the paper.
>
> Additionally, you pointed out
>
> > The authors first defined $Z$ in section 2.2 but it first appears in section 2.1.
>
> We agree with the reviewer that introducing $Z$ in Section 2.2 was confusing, and we simply remove it from Section 2.2 in the updated version of the paper. Now, a generic sequence of observations $Z_1, Z_2, \dots$ appears only in Section 2.1 where it is used to formally define a sequential test for unfamiliar readers.
>
> You asked
>
> > the definition of $R^{(t)}(f)$, what is the expectation over?
>
> After relaxing the i.i.d. assumption, for each data point on the target domain $(X'_i,Y'_i)$, $i=1,2,\dots$ the expectation is taken over the corresponding distribution. We clarify that point in the definition of the running risk $R^{(t)}(f)$.

---

### Official Review · Reviewer_9YNK · 2021-10-26

**Correctness:** 4
**Technical Novelty And Significance:** 4
**Empirical Novelty And Significance:** 3
**Recommendation:** 8
**Confidence:** 3

**Details Of Ethics Concerns:**

None.

**Main Review:**

The proposed framework leverages well-established results in statistics to detect dataset shifts. While the framework addresses the problem in the most practical settings (sequential), the points of comparison with the existing frameworks were not made rigorous enough for the reader. Thus, I have the following questions -

1. Why is sequential testing necessary in dataset shift under i.i.d assumption? I can imagine it to be useful under the independence assumption. Under the i.i.d setting, how does the performance of the proposed framework compare to Kamulete, 2021? Is the power of the test comparable? Can a similar curve (Figures 2,3,4)  for a number of samples be obtained for Kamulete, 2021? When will the testing framework of Kamulete, 2021 fail or work better? MNIST-C experiments are under the i.i.d dataset shift. Can a comparison with this testing framework be made?

2. Similarly, when the i.i.d assumption is violated (distribution drift), can the comparison with the testing framework of Vovk et al. 2021 be made?


Minor comments -
- Figure 4(a): Horizontal line is yellow, not red
- PM and CM are not spelt out in the main paper
- “the this” often used in the paper
- “ intervene in a data-adaptive way”: Does this refer to model retraining? Are there other examples?


**Summary Of The Paper:**

The authors have proposed a framework for detecting dataset shift (under i.i.d assumption or via drift) such that the model’s performance suffers a significant (defined by the user) performance degrade. The framework accommodates several types of risk functions that can be used to evaluate the model’s performance. The underlying engine of the framework is sequential testing via non-parametric sequential estimation. Such a framework can potentially be useful for raising alarms or indicating when the model needs to be retrained.

**Summary Of The Review:**

Based on my knowledge and background research conducted while reviewing this work, I think the authors have proposed a novel framework to detect dataset shifts. It can potentially be used to improve several applications fraught with predictive models. The work is well supported with the help of illustrative examples. My only concern is their comparison with the existing works on dataset shift detection is not well explained/not supported.

---

> ### Author Response · Authors · 2021-11-21
> **Response to Reviewer 9YNK (1/2)**
>
> We warmly thank the reviewer for constructive and actionable feedback on the paper. Below, we address certain points that have been raised in the review. We hope that the reviewer may consider increasing their score if many/most of the concerns were reasonably addressed.
>
> You asked
>
> > **(joint answer)** (a) Why is sequential testing necessary in dataset shift under i.i.d assumption? I can imagine it to be useful under the independence assumption. Under the i.i.d setting, how does the performance of the proposed framework compare to Kamulete, 2021? Is the power of the test comparable? Can a similar curve (Figures 2,3,4) for a number of samples be obtained for Kamulete, 2021? When will the framework of Kamulete, 2021 fail or work better? Can a comparison be made? (b) When the i.i.d assumption is violated, can the comparison with Vovk et al. 2021 be made?
>
> We first want to point out that Kamulete, 2021 uploaded a preprint to arXiv a few weeks before the ICLR deadline. We made the conscious choice to mention the paper, but it seems to be held against us --- in our view, it seems a bit unfair to ask us to explicitly compare to their work at short notice, especially since it is unpublished and hence can hardly be considered the standard to which our work must be held to, and also because they provide no code for their work. In contrast, Vovk et al 2021 was published earlier this year, so that comparison is indeed fair to ask about. Nevertheless, we will answer all the questions of the reviewer below.
>
> (a) A common assumption for covariate shift papers is to assume that the source data is iid from $P$, while the target data is iid from $Q \neq P$, the latter is what we refer to as the iid assumption. A more general setting is to assume that the target data are independent, but not identically distributed from a single distribution (one can think about it as a draw from $Q_1 \times Q_2 \times \dots $). When one tests for (harmful) distribution shift, one can either make the iid (from $Q$) assumption, or avoid it, and we develop tests in either case. Of course, the latter setting is more general, but the former setting is common in the literature, so we handle both. In both settings, the shifts/drifts may be benign or harmful, so the question we handle is still relevant.
>
> While the method described in Kamulete, 2021 is an important contribution to the field, it represents a fixed-time test (that can be applied only once at a predecided sample size or time, like one year from today), and thus it is simply not valid under the sequential setting considered in our work, where we are allowed to monitor the performance of a deployed model every hour of every day in the upcoming year. To illustrate the shortcomings of deploying fixed-time tests under continuous monitoring, we performed a series of new experiments that are presented in Appendix A.1 of the updated version of the paper.
>
>  (b) Vovk et al., 2021 use conformal test martingales that do not compare the performance of a model on the source and target domains, and thus fail to account for the malignancy of a shift in an interpretable way. Such martingales are primarily used to detect *any* deviations from the i.i.d. assumption. Further, their procedure crucially depends on choosing a nonconformity score that underlies a conformal p-value. To the best of our knowledge, this and some other design choices still require more study for the approach to become fully practically relevant. We have included a detailed study of conformal test martingales in Appendix A.2 of the updated version of the paper. Like us, they also require access to true labels. The main shortcoming of deploying conformal test martingales (CTMs) include:
>
> - **Possibly low power:** only extreme (severe and sharp) shifts/drifts are consistently detected.
>
> - **Poor differentiation between harmful and benign cases:** it is hard to incorporate evaluation of the malignancy of a change in an interpretable way.
>
> - **Saturation:** values of CTMs (larger value implies more evidence for a shift) could saturate and then start to decrease after some time. This is because they measure deviations from iid data, not degradations in performance from some benchmark, like source accuracy; after some time, the shifted distribution becomes the ''new normal'', and is no longer treated as a shift by the method, even if the accuracy is now worse than it was at the start.

---

> ### Author Response · Authors · 2021-11-21
> **Response to Reviewer 9YNK (2/2)**
>
> You mentioned
>
> > **(joint answer)** (a) Figure 4(a): Horizontal line is yellow, not red. (b) PM and CM are not spelt out, (c) ''the this'' is often used
>
> We thank the reviewer for catching the typos. We have fixed the aforementioned issues.
>
> You asked
>
> > ''intervene in a data-adaptive way'': Does this refer to model retraining? Are there other examples?
>
> Possible interventions are not limited only to model retraining. In the Introduction, we also comment on performing predictive uncertainty quantification. Certain procedures that are referenced in the work can handle two particular forms of distribution shifts (label and covariate shifts) in a post-hoc manner, that is without retraining the original model. Another possible intervention could also involve the incorporation of time-series models, rather than simply retraining a neural net as if it is a different iid dataset. We are agnostic to the kind of intervention needed, and focus on when to raise an alarm.

---

> ### Comment · Reviewer_9YNK · 2021-11-28
> **Thanks for the clarifications!**
>
> I have gone through the author's response. I agree that comparison with Kamulete et al. is not a fair one to ask. Nevertheless, the authors have performed necessary experiments to bring out the points of difference between their proposal and the other two testing frameworks. I reckon that this will greatly help the readers in their understanding. I am convinced that the utility of the proposed testing framework will be widely applicable to practical settings.
>
> I have no further questions. I have also increased my scores. I thank the authors for their efforts.

---

### Official Review · Reviewer_oNqc · 2021-11-05

**Correctness:** 4
**Technical Novelty And Significance:** 2
**Empirical Novelty And Significance:** 2
**Recommendation:** 6
**Confidence:** 3

**Main Review:**

$\textbf{Strengths}$

Detailed and clear with proper explanations and proofs of the presented schema and related propositions.

Builds on related work in the field with (to the best of my knowledge) a novel contribution.

Discussion of relevant distribution shifts and their effect on classifiers is really interesting (see weaknesses).

$\textbf{Weaknesses}$

Minor grammatical/style issues. For example, on page 3 the $\textbf{Multiclass losses}$ sub-header has a period, but other similar ones do not. I suggest adding a period to all of them for consistency. There are also other grammatical issues such as on page 3: "those naturally arise in multilabel classification, where more than a single label can be the correct one, or as a result of post-processing point predictors, which could target...." where "those" is a vague pronoun reference and the use of commas is incorrect.

Although the contribution is novel (to my knowledge), I think that the paper is missing an explicit discussion as to why this contribution is important (e.g. a particular case where this new schema enables users to prevent something that they otherwise could not and being unable to has negative repercussions).

Although the discussion of relevant distribution shifts and their effect on classifiers is really interesting, it left me wanting more. I think the paper could benefit from expressing how the testing schema presented can be used to identify malignant shifts and how to do so (e.g. humans thing translation wouldn't be an issue but it is, how do we discover more facts like this provable?).

**Summary Of The Paper:**

This paper proposes a sequential testing schema for identifying malignant distribution shifts that indicate a necessary update such as retraining. The framework presented is designed to be generalizable and applicable to many scenarios (binary, multiclass, set functions, etc.).

**Summary Of The Review:**

I give the paper a 6 because I find the contribution novel and clear and would be happy to see it accepted in the conference; however, I would not champion pushing to accept given my belief that there is room for significant improvement.

---

> ### Author Response · Authors · 2021-11-21
> **Response to Reviewer oNqc**
>
> We warmly thank the reviewer for constructive and actionable feedback on the paper. Below, we address certain points that have been raised in the review. We hope that the reviewer may consider increasing their score if many/most of the concerns were reasonably addressed.
>
> You mentioned
>
> > Minor grammatical/style issues. For example, on page 3 the sub-header has a period, but other similar ones do not... There are also other grammatical issues such as on page 3... where ''those'' is a vague pronoun reference and the use of commas is incorrect.
>
> We thank the reviewer for pointing these English issues out. We have updated the manuscript and the feedback from the reviews has allowed us to improve our presentation of the work.
>
> You also mentioned
>
> > **(joint answer)** (a) Although the contribution is novel (to my knowledge), I think that the paper is missing an explicit discussion as to why this contribution is important ... (b) I think the paper could benefit from expressing how the testing schema presented can be used to identify malignant shifts and how to do so...
>
> We have updated the writing in a way that hopefully addresses the aforementioned concerns.
>
> - (a) We now cleanly state the contributions of the current work and possible negative repercussions of naive testing in the Introduction; to summarize, our methodology allows for continuously monitoring a deployed model and it raises an alarm only when a harmful (not arbitrary) distribution shift is detected, with a guarantee that if there is no harmful shift, then with high probability it will never raise an alarm (its false alarm rate is provably controlled, even with a drifting target distribution, as long as the target risk stays close to the source risk). Further, we have performed a series of additional experiments that illustrate the shortcoming of deploying traditional (fixed-time) tests under continuous monitoring. The corresponding analysis can be found in Appendix A.1 of the updated version of the manuscript.
>
> - (b) This is more straightforward: when you deploy the model, also run our sequential test in parallel; if our test rejects the null, we have detected a malignant shift (and the user must then make a choice on what action to take next, the simplest option being retraining).

---

### Official Review · Reviewer_mEfm · 2021-11-07

**Correctness:** 3
**Technical Novelty And Significance:** 2
**Empirical Novelty And Significance:** 2
**Recommendation:** 6
**Confidence:** 3

**Main Review:**

Strengths: The paper applies sequential testing to detect harmful distribution shifts. The setting is quite different from existing work.

Weaknesses:
Writing. The writing quality and the presentation of the paper can be substantially improved. For example, in section 2, It might be better to move the loss functions to the experimental section, as they are not part of the algorithm.

Novelty. Overall, I don't see enough originality, and it's not very challenging to adapt the sequential testing to this setting. The main idea is to detect the bad distribution shifts by nonparametric sequential testing based on comparing the risk function on the target data and the risk function on the source data. The monitoring statistics are the risk functions (and their CI).

Comments. 1. The proposed method requires the labels from the test data, which is a restrictive setting for practical use. The paper claims that the label can be revealed in a delayed fashion, but we could also do batch detection for the past observations (and wait for a period and redo the testing). The paper doesn't provide any expected sample size analysis. I think it'd be better to give some comparison with the batch detection algorithm.  2. Since the algorithm requires to compute the CI at each time step, how efficient is it?

**Summary Of The Paper:**

This paper develops a tool for testing online whether the performance of a model on the test data becomes significantly worse than the performance on the training data, which allows differentiating between benign and harmful shifts. The proposed framework is based on sequential testing for a significant risk increase. The theoretical guarantee of the type I error control is provided. The paper also gives simulation and real data experiments to demonstrate the efficacy of the proposed algorithm.

**Summary Of The Review:**

I'd recommend reject at this point.

-----------------------------------------------------
Thank the authors for the response! The authors substantially improved the writing and the presentation in the updated version. They also added new content regarding comparison with other papers/other potential methods, which makes the contribution more clear. I read reviews from other reviewers as well as the corresponding responses. The answer clears most of my concerns. Although I think the theoretical contribution is not significant, adapting sequential estimation to the practical setting would be nice.

---

> ### Author Response · Authors · 2021-11-21
> **Response to Reviewer mEfm (1/2)**
>
> We warmly thank the reviewer for constructive and actionable feedback on the paper. Below, we address certain points that have been raised in the review. We hope that the reviewer may consider increasing their score if many/most of the concerns were reasonably addressed.
>
> You mentioned
>
> > The writing quality and the presentation of the paper can be substantially improved. For example, in section 2, It might be better to move the loss functions to the experimental section.
>
> We have updated the writing incorporating the feedback from the reviews, and hope that it allowed to improve the writing quality and the presentation of the paper. In particular, discussion of the relevant loss functions has been moved to the Appendix with proper referencing in the main paper.
>
> You raised the following concern
>
> > Novelty. Overall, I don't see enough originality, and it's not very challenging to adapt the sequential testing to this setting. The main idea is to detect the bad distribution shifts by nonparametric sequential testing based on comparing the risk function on the target data and the risk function on the source data.
>
> We kindly disagree with the reviewer regarding the novelty of the contributions since, to the best of our knowledge, the results presented in this work have not been established before. While the current work relies heavily on *recent* advances in different areas, e.g., construction of the confidence sequences, the corresponding application to testing for harmful distribution shifts is new, and important, since this is not currently being employed in the tech industry. However, we agree that outlining the main contributions and distinguishing presented results from the ones available before in the literature had to be improved. We tried to accomplish that in the updated version of the manuscript.
>
> You wrote
>
> > The proposed method requires the labels from the test data, which is a restrictive setting for practical use. The paper claims that the label can be revealed in a delayed fashion, but we could also do batch detection for the past observations (and wait for a period and redo the testing).
>
> We agree that requiring observing labels represents a limitation of the proposed framework which we highlighted in the paper. However, we kindly disagree that batch detection represents a compelling alternative to the proposed framework. Redoing traditional (fixed-time) testing every week (say) requires performing corrections for multiple testing, as otherwise the validity of the procedure is lost. If corrections are applied (for example using a Bonferroni correction, ie union bound), then the resulting procedure becomes too conservative, due to the dependence among the tests being ignored (and not being taken advantage of). However, performing testing via sequential estimation allows to accumulate evidence over time, without throwing away any data or the necessity of performing corrections for multiple testing (these are implicitly handled efficiently using martingale methods that are underlying the sequential estimation procedures).
>
> We have added a note to the main paper describing the issues with doing traditional (fixed-time) testing, with or without a correction. Further, we have performed a series of additional experiments that illustrate the shortcoming of deploying fixed-time tests under continuous monitoring. The corresponding analysis can be found in Appendix A.1 of the updated version of the manuscript.

---

> ### Author Response · Authors · 2021-11-21
> **Response to Reviewer mEfm (2/2)**
>
> You mentioned
>
> > The paper doesn't provide any expected sample size analysis. I think it'd be better to give some comparison with the batch detection algorithm.
>
> This is a good but hard question. We remark here that our problem setup is highly nonparametric: the distribution of $P(X,Y)$ under the source, as well as the sequence of target distributions (one for each test point $X_i, Y_i$) are completely unspecified and may be heavy-tailed and varying in arbitrary manners. Most of the sequential analysis literature from Wald (1940s) has been very parametric in nature. Nonparametric testing is, by contrast, relatively young, and much less progress has been made because of less structure (like exponential families) to take advantage of.
>
> Unfortunately, while proceeding under a general nonparametric setting allows for developing a more universal testing framework as no parametric assumptions are made (and thus it can be deployed tomorrow at any IT company), it makes that theoretical analysis of the expected sample size a highly nontrivial task. Regarding the empirical side, we perform a detailed study of the performance of the framework under a number of settings. The key advantage of the proposed framework is validity under adaptive data collection scenarios when batch detection algorithms are simply not valid --- if we knew at time 0 (when we deploy the model) that running a test after observing a target sample size of 250 would detect a harmful shift, we would certainly do that, but how would we possibly know at time 0 at what time the harmful shift begins, how strong the effect is, and at what sample size we would have enough power to detect the effect? Thus, from our viewpoint, batch tests are not a relevant benchmark to compare to, because nobody should be continuously monitoring a batch test (they would be inundated with false alarms) and otherwise they would have to fix a sample size ahead of time to run the test, while this may either miss the effect if it is too small, or be way too late to detect the effect (compared to the sequential test) if the sample size is too large. We hope that the updated writing and the additional empirical study presented in Appendix A.1 (performance of fixed-time tests under continuous monitoring) reflect these points in a clearer way.
>
> You also asked
>
> > Since the algorithm requires to compute the CI at each time step, how efficient is it?
>
> The complexity of running the proposed framework reduces to computing confidence sequences, which itself can be performed efficiently (a constant time update to the CI at time $t$ yields the CI at time $t+1$). One possibility for further computational improvements is stated in Remark 1: one does not have to recompute the confidence sequences after each new label is collected, but instead, those can be re-estimated after processing minibatches of data.

---

### Official Review · Reviewer_P77d · 2021-11-08

**Correctness:** 4
**Technical Novelty And Significance:** 2
**Empirical Novelty And Significance:** 4
**Recommendation:** 8
**Confidence:** 4

**Main Review:**

The work highlights an important problem of alerting when model performance differs from that in training, statistically significantly. The framework is broadly applicable to any model and multiple loss functions. Multiple issues in monitoring of models are addressed such as distribution shifts, sequential testing, different loss function. The solution uses some recent work on non-parametric, non-asymptotic testing in a novel way and has good performance under comprehensive set of experiments.

The main weakness is the presentation. The resolution of issues with past work, such as shifts and sequential testing, is not described well. Technical details on computing the confidence bounds are missing. Apart from these details, the framework and the experiments are described well.

---

## Questions to address in the response

1. What is the difference in the method between the two settings i.i.d. and independent test examples?
2. What type of distribution shifts such as covariate, label shift can be addressed?
3. Please comment on the utility of testing the running risk? As the model decisions have already been made, the running risk tells the risk in the (recent) past and not the imminent risk. If the distribution shifts considerably, then the imminent risk can be considerably different. In what scenarios (or some smoothness assumptions on the drift) can such a test for running risk help in giving early warnings? I feel elaborating on this detail gives a better understanding on when to use such methods.
4. What can be included to describe the method in more detail? Please see suggestions below.
5. Does the alternate methods (EB and Hoeffding based ones) provide valid intervals in the considered setting of shifts and sequential testing? How would we expect such methods to perform, too loose or incorrect confidence intervals?
6. How to choose between betting and conjugate mixture empirical-Bernstein methods for confidence bounds for source and target risk? Both are used in real data experiments, one for source and one for target. My understanding is both provide symmetric intervals and thus are applicable.

---

## Suggestions

The use of time-uniform confidence sequences to resolve issues with past works should be explained in more detail for readers not familiar with advantages of such confidence sequences.

The description of the exact testing procedures used needs to be self-contained including how to find upper and lower confidence bounds for the risk in terms of a given model, loss, and confidence parameter \delta.

The context for introducing different loss functions at the start of Section 2 is not clear from the Introduction, which mostly talks about sequential testing without much attention to different loss functions. The discussion on loss functions is not related back to the problem or the method.

Please add the takeaway from Figure 4b. Should we expect other methods to take more samples to detect the harmful shift?

Results for Figure 4a seems to show that the betting based bounds are tighter only for small number of samples and about the same for otherwise. The discussion on the Figure can be made more specific, instead of saying that it is tighter in general.

The contributions of the work can be outlined at the end of Introduction so that reader has some context.

The necessity of time-uniform confidence sequences can be elaborated for readers not familiar with multiple testing under dependent data. Consider highlighting the issue with using traditional tests.

Please briefly describe what is \beta used in Section 3.2 to output prediction sets in the main text. It is fine to refer to the text in the Appendix for details.

Clarify if the method works for any loss function or the theoretical results need some conditions on the loss function.

Describe the full-form of PM-EB and the formulae for the bound.

In the expression for running risk before Eq (6), clarify what is the distribution for the expectation.

It will be worth mentioning (if not already) that instead of comparing against the upper confidence bound for source risk, one can use a given risk threshold deemed to be harmful and the method still applies.

The line is disconnected from the neighbouring sentences - If accuracy is a guiding performance metric, one would rarely deploy models performing only sightly better than prediction by chance.

**Summary Of The Paper:**

The work presents a framework for alerting when the risk of a deployed model exceeds pre-defined levels. Observing a sequence of target instances, coming from possibly different distribution than source one, it defines two hypothesis tests (based on additive and multiplicative error) for harmful changes to the risk. It reduces the problem to sequential estimation of bounds for the target risk and uses recent work on time-uniform confidence sequences to design valid hypothesis testing methods. Empirical performance on a controlled synthetic setting and real world data from MNIST, CIFAR show that the methods detects harmful changes to risk in reasonable samples.

**Summary Of The Review:**

The work addresses an important problem for monitoring models after deployment for harmful changes and derives principled hypothesis tests. Empirical study sufficiently validates the method on real datasets. My main concern is the presentation of the method. It lacks details on related to implementation and guarantees for the methods for readers who are unfamiliar with past work (Waudby-Smith and Ramdas 2021, and Howard et al. 2021). The work provides provides a strong contribution and I feel, subject to the response, the presentation can be improved.

---
Based on the response

The presentation of the methods has been substantially improved. In particular, the contributions paragraph at the end of Introduction gives a good overview of the work. I would encourage authors to more prominently list the confidence bound variants (betting-based, PM-H, PM-EB, CM-EB, Hoeffding) in Sec 2.2 to more clearly layout the options to implement the tests.

---

> ### Author Response · Authors · 2021-11-21
> **Response to Reviewer P77d (1/3)**
>
> We warmly thank the reviewer for constructive and actionable feedback on the paper. We hope that the updated writing and filling in the missing components improve the overall presentation of the paper, and allow for distinguishing the contributions of the current work among the methods developed earlier. Below, we address in detail several points that have been raised in the review.  We hope that the reviewer may consider increasing their score if many/most of the concerns were reasonably addressed.
>
> You asked
>
> > What is the difference in the method between the two settings i.i.d. and independent test examples?
>
> In the second case, the examples are independent but not identically distributed (and the form of the drift in the distribution of the data could change with time and depend on the past observations).
>  From the technical perspective, the difference arises in which particular concentration result is applicable for a particular setting. While the betting-based confidence sequence necessitates assuming that random variables share a common mean, the approach based on conjugate-mixture empirical-Bernstein allows us to lift this assumption and handle a varying mean. We have updated the writing to reflect this distinction.
>
> You also asked
>
> > What type of distribution shifts such as covariate, label shift can be addressed?
>
> It applies to settings with covariate shift, or label shift, or any other general type of shift. One of the main advantages of the proposed framework is its flexibility concerning the form of the underlying shift. It does not require constraining the possible types of shifts or drifts, and in general, is also applicable for different mixtures of shifts.
>
> You mentioned
>
> > Please comment on the utility of testing the running risk? As the model decisions have already been made, the running risk tells the risk in the (recent) past and not the imminent risk. If the distribution shifts considerably, then the imminent risk can be considerably different. In what scenarios (or some smoothness assumptions on the drift) can such a test for running risk help in giving early warnings?
>
> This is a good and natural question of practical importance. At this point, our framework simply raises an alarm (with a guarantee on the low probability of a false alarm despite continuous monitoring) and it effectively asks a human to intervene or automatic retraining to occur because it has observed a run of data which was different from the training data in a harmful way. It cannot and does not take a stance on whether such a drift will continue into the future; this would be a natural conjecture, but the plausibility of that conjecture would depend on the domain, and we prefer to leave that judgment to the maintainer of this system (which as constructed, could be almost any deployed ML system in any scientific or industry context, making it hard for us to say more in any generality). That said, intuitively speaking, it seems plausible that if the data were smoothly drifting, and that drift has worsened the risk, then continuing to deploy the model could continue to worsen the risk. Such features can be easily added onto the system if needed, for example extrapolating a smooth risk curve into the future to guesstimate the imminent risk, and is complementary to the construction of our system itself.

---

> ### Author Response · Authors · 2021-11-21
> **Response to Reviewer P77d (2/3)**
>
> You wrote
>
> > **(joint answer)** (a) The use of time-uniform confidence sequences to resolve issues with past works should be explained in more detail for readers not familiar with advantages of such confidence sequences. (b) The necessity of time-uniform confidence sequences can be elaborated for readers not familiar with multiple testing under dependent data. Consider highlighting the issue with using traditional tests. (c) Does the alternate methods (EB and Hoeffding based ones) provide valid intervals in the considered setting of shifts and sequential testing? How would we expect such methods to perform, too loose or incorrect confidence intervals?
>
> - (a-b) Adapting traditional (fixed-time) testing to the sequential setting would require performing corrections for multiple testing (ie multiple views of the data, at different timepoints); if the correction is not performed, validity is lost (false positives will abound) but if such a correction is performed, the test will be too conservative, due to the dependence among the tests being ignored (and not being taken advantage of). However, performing testing via sequential estimation allows to accumulate evidence over time, without throwing away any data or the necessity of performing corrections for multiple testing (these are implicitly handled efficiently using martingale methods that are underlying the sequential estimation procedures). We have added a note to the main paper describing the issues with doing traditional (fixed-time) testing, with or without a correction. Further, we have performed a series of additional experiments that illustrate the shortcoming of deploying fixed-time tests under continuous monitoring. The corresponding analysis can be found in Appendix A.1 of the updated version of the manuscript.
>
> - (c) We presume the question refers to *our* EB and Hoeffding based confidence sequences, and not the classical EB/Hoeffding confidence intervals. Our methods do provide valid intervals (for the running true risk) in the presence of arbitrary shift, even at stopping times, and can be used for testing (by seeing if the intervals fail to contain the training risk). The latter, classical, methods are not sequentially valid as is, and if corrections for multiple testing are performed, they would be too conservative.
>
> You commented
>
> > **(joint answer)** (a) description of testing procedures needs to be self-contained... (b) How to choose between betting and conjugate mixture empirical-Bernstein methods for confidence bounds for source and target risk?...
>
> - (a) We have updated the writing in order to state the procedures cleaner.
>
> - (b) Choosing the procedure for sequential estimation is motivated by the set of assumptions one proceeds with (under which the testing procedure provides valid inference). For example, using the betting-based confidence sequence requires making the common-mean assumption (ie the target distribution may be different from the source, but does not change over time, meaning that there is a shift but no drift), while conjugate mixture empirical-Bernstein works in the presence of any drift/shift and requires no common-mean assumption. We have updated the writing to make it clearer. Since the common-mean assumption is reasonable for the source distribution (for which we have all the points), the former suffices. But for the target data, we use the latter to avoid making any assumption (if there was already a distribution shift from training to deployment, there may also be a further drift over time).
>
> You mentioned
>
> > **(joint answer)** (a) The contributions can be outlined at the end of Introduction for context. (b) The context for introducing different loss functions at the start of Section 2 is not clear... (c) Clarify if the method works for any loss function or the theoretical results need some conditions.
>
> - (a) The main contributions are outlined in the updated version of the paper.
> - (b) We have moved the discussion of relevant losses to the Appendix with proper referencing in the main paper. Now, the second section of the paper focuses purely on sequential testing.
> - (c) We have clarified that our results work for any *bounded* loss function. For example, the unbounded log-loss could cause wild fluctuations (say negative infinity), and thus it is hard to estimate the true risk in terms of the empirical risk; this is a broader issue that applies outside of this paper (even in iid settings).
>
> You also mentioned
>
> > **(joint answer)** (a) Please add the takeaway from Figure 4b. Should we expect other methods to take more samples to detect the harmful shift? (b) Results for Figure 4a seem to show that the betting-based bounds are tighter only for a small number of samples and about the same for otherwise. The discussion on the Figure can be made more specific, instead of saying that it is tighter in general.
>
> We have updated the captions to Figures 4a and 4b stating the takeaway messages clearer.

---

> ### Author Response · Authors · 2021-11-21
> **Response to Reviewer P77d (3/3)**
>
> You mentioned
>
> > Please briefly describe what is $\beta$ used in Section 3.2 to output prediction sets in the main text. It is fine to refer to the text in the Appendix for details.
>
> We have removed the notation $\beta$ from the main paper to avoid confusion. Now, a detailed discussion of the set-valued predictor construction is presented in the Appendix, and it is properly referenced in the main paper.
>
> You wrote
>
> > Describe the full-form of PM-EB and the formulae for the bound.
>
> We have updated the Appendix which now contains all relevant details regarding the concentration results considered in the paper.
>
> You asked
>
> >In the expression for running risk before Eq (6), clarify what is the distribution for the expectation.
>
> After relaxing the i.i.d. assumption, for each data point on the target domain $(X'_i,Y'_i)$, $i=1,2,\dots$ the expectation is taken over the corresponding distribution. We clarify that point in the definition of the running risk $R^{(t)}(f)$.
>
> You also mentioned
>
> > It will be worth mentioning (if not already) that instead of comparing against the upper confidence bound for source risk, one can use a given risk threshold deemed to be harmful and the method still applies.
>
> We thank the reviewer for pointing that option (compelling for the cases of misclassification/miscoverage risks). The corresponding note has been added in the updated version of the manuscript.
>
> You wrote
>
> > line is disconnected from the neighboring sentences --- ''If accuracy is a guiding performance metric...''
>
> The corresponding part of the paper has been rephrased for better clarity.

---

### Author Response · Authors · 2021-11-21
**Paper has been revised incorporating the received feedback**

Dear reviewers and area chair,

We thank you for the comments and suggestions on improving the current work. We are committed to addressing all the concerns that have been raised in the reviews.

To make it easier to navigate through the updated version of the paper, we highlighted **major** updates with **blue color**. If the paper is accepted, the highlighting will be removed for the camera-ready version.

---

### Decision · Program_Chairs · 2022-01-20

**Decision:**

Accept (Poster)

**Comment:**

This paper is concerned with the problem of distribution shift, and develops techniques for detecting when the risk of a deployed model performs significantly worse on a testing distribution than on the training distribution.

The reviews for this paper were extremely consistent: after the discussion period, all five reviewers unanimously recommended acceptance, and several praised the authors for significantly improving their paper in response to reviewer criticism. Outstanding issues are (i) motivating the importance of the setting, and (ii) comparing with prior work. None of the reviewers seemed to think that these issues should be barriers to acceptance, but please seriously consider them, and all reviewer concerns.